# EMERGENT MIXTURE-OF-EXPERTS: CAN DENSE PRE-TRAINED TRANSFORMERS BENEFIT FROM EMERGENT MODULAR STRUCTURES?

## ABSTRACT

Modular neural architectures demonstrate superior generalization performance. Existing modular neural networks are generally *explicit*. Their modular architectures are pre-defined with individual modules expected to implement distinct functions. Conversely, recent works reveal that there exist *implicit* modularity in standard pre-trained transformers, namely *Emergent Modularity*. They indicate that such modular structures exhibit during the early pre-training phase and are totally spontaneous. However, most pre-trained transformers are still treated as monolithic models in the pre-train-and-fine-tune paradigm with their modular natures underutilized. In this work, we explore *whether and how leveraging emergent modularity during fin-tuning could bring better generalization*. We showcase that standard pre-trained transformers could be fine-tuned as their sparse Mixture-of-Expert (MoEs) counterparts without introducing any extra parameters. Such MoEs is derived from the emergent modularity and is referred to emergent MoE (EMoE). Extensive experiments (we tune 1785 models) on vision and language downstream tasks and models (22M to 1.5B) demonstrate that fine-tuning EMoE effectively improves in-domain and out-of-domain generalization compared with vanilla fine-tuning. Our analysis shows that EMoE could mitigate negative transfer during fine-tuning. Further ablations suggest that EMoE is robust to various configurations and can scale up to very large language models (e.g., Llama-30B).

## 1 INTRODUCTION

Modularity attracts considerable attention from the artificial intelligence community (Auda & Kamel, 1999). Neural networks with modular designs, termed Modular Neural Networks (MNN), have exhibited a wide range of advantages, including adaptation (Shen et al., 2023b),data efficiency (Bengio et al., 2020), and better generalization abilities (Goyal & Bengio, 2020; Weiss et al., 2022),. Typical MNNs are usually *explicitly* modular. Their modular structure is pre-defined and they are expected to achieve a divide-and-conquer solution for the given task. For example, Andreas et al. (2016) design separate visual and language functional modules and jointly train them to solve visual question-answering tasks. Among various MNNs, Mixture-of-Experts (MoEs) employ a conditional computation strategy where different submodules - so-called experts - are expected to be activated by different types of inputs. MoEs see substantial success in various domains (Shen et al., 2023a; Chen et al., 2023b; Mustafa et al., 2022; Bao et al., 2022) in the era of large-scale transformers, and therefore becomes a widespread modular neural architecture.

Apart from *explicit* MNN, Csordás et al. (2021); Agarwala et al. (2021) study whether standard neural networks become *implicitly* modular after training and discover spontaneously emerged modular structure in small-scale CNNs and LSTMs. For more complicated large-scale pre-trained transformers, initial observations (Zhang et al., 2022b; Li et al., 2022) reveal notable sparse activation patterns within the Feed-Forward Networks (FFNs). Specifically, they find that in T5-Base (Raffel et al., 2020) and ViT-B16, only 3.0% and 6.3% neurons are activated during one forward process. However, sparsity does not imply modularity. Sparsely activated FFNs neurons do not ensure to have remarkable functional division. Therefore, Zhang et al. (2023) further utilize handpicked semantic and knowledge-intensive tasks to probe the nature of neurons in FFNs. They observe a strong correlation between activation and specific tasks, further discovering clear function-based neuron grouping of the model and summarizing such phenomenon as *Emergent Modularity (EM)*.

However, pre-trained transformers are generally treated as monolithic models in the current standard pre-train-and-fine-tune paradigm. We wonder whether their EM and the potential improvements brought by EM are ignored. In this work, we narrow it down to the following specific question: whether introducing EM could be conducive to models' downstream generalization ability. We thus employ a method from (Zhang et al., 2022b) to derive the Emergent MoE (EMoE) based on the EM of original pre-trained transformer. We compare the fine-tuning performance of EMoE and the original model to answer our research question. We validate our conclusions in a wide range of configurations: (1) tasks from various modalities: vision tasks (Gulrajani & Lopez-Paz, 2021)and language tasks (Wang et al., 2019b; Yang et al., 2023); (2) different pre-trained transformers: ViT (Dosovitskiy et al., 2021), BERT (Devlin et al., 2019), and GPT2 (Radford et al., 2019), ranging from 22M to 1.5B parameters. (3) different evaluation settings: in-domain (ID) evaluation and OOD evaluation; (4) different fine-tuning methods: full fine-tuning and LoRA tuning (Hu et al., 2022).

Our experimental results are summarized in Figure 1. We find that **fine-tuning EMoE achieves better generalization performance** than vanilla fine-tuning and competitive performance with one strong baseline GMoE (Li et al., 2023). The improved downstream performance is also observed when being applied to multi-task learning and very large language models. Our analysis indicates that the reason for improvements is that **EMoE could deactivate neurons with negative transfer effects during fine-tuning**. Meanwhile, our ablation studies indicate the EMoE's insensitivity to various hyper-parameter configurations. We hope our practice and research discoveries can serve as an example attempt towards further utilizing the EM of pre-trained transformers.

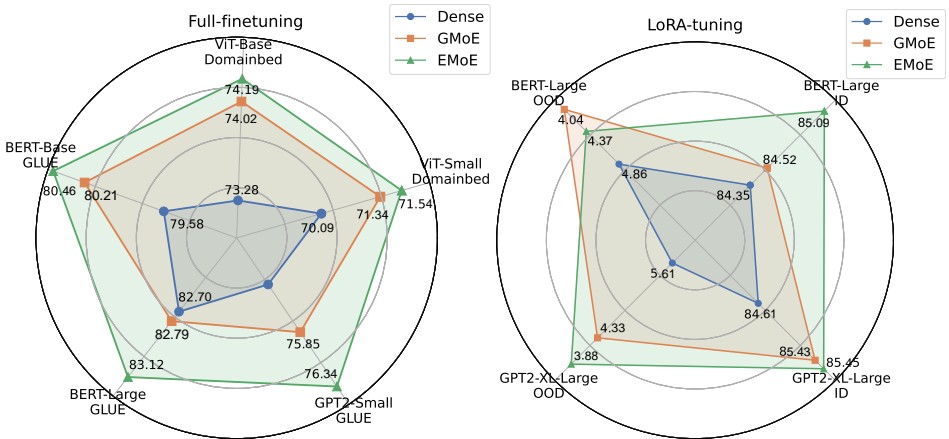

Figure 1: EMoE achieves stronger ID and OOD performances compared with baselines in both full-fine-tuning (Left, average accuracy) and LoRA tuning settings (Hu et al., 2022) (Right, average accuracy for ID and Friedman rank (Friedman, 1940) for OOD). Results are summarized from Section 4.

## 2 RELATED WORK

Leveraging modular designing into neural networks has various advantages, including interpretability (Pfeiffer et al., 2023), scalability (Chowdhery et al., 2022), multi-task learning abilities (Chen et al., 2023a), and OOD generalization abilities (Goyal et al., 2021; Li et al., 2023). MoEs (Szymanski & Lemmon, 1993) is currently a standard framework for developing modular neural networks (Shen et al., 2023b; Shazeer et al., 2017; Fedus et al., 2022; Zhang et al., 2022a). Despite explicit modular neural nets, Hod et al. (2021); Csordás et al. (2021) explore emergent modular structures in CNNs and LSTMs. Some recent works (Zhang et al., 2022b; Li et al., 2022) focus on the sparsity of more complicated pre-trained transformers. Based on their observations, Zhang et al. (2023) recently explore modularity in pre-trained transformer FFNs by employing handpicked semantic and knowledge-intensive tasks to probe the modular nature of pre-trained transformers.

Works most related to our work are those that utilize off-the-shelf pre-trained models to construct MoEs. For example, GMoE (Li et al., 2023) and Upcycling (Komatsuzaki et al., 2023) copy the FFNs from a trained transformer model to form the MoEs architecture. Their modular structure is introduced by replicating existing FFNs modules, leaving EM within pre-trained FFNs unexplored. We choose GMoE as one of our baselines. MoEfication (Zhang et al., 2022b) explores the EM within

the model. They seek to improve the inference efficiency by replacing the original FFNs layer with MoEs. MoEfication utilizes sparse activations to reduce inference overhead within FFNs and does not touch on how EM influences the training stage. Although more sophisticated methods could be developed, we adopt one simple method in the ablation studies of MoEfication paper (Zhang et al., 2022a) to externalize EM from the original model: clustering-based expert construction and avg-k gating. We empirically find that such a simple method can validate the improvements brought by EM and answer our research questions. We thus leave more elaborated methods for future works. Please refer to the Appendix A.2 for a detailed comparison of EMoE with related works.

## 3 METHODOLOGY

### 3.1 PRELIMINARIES

**Transformer Feed-Forward Networks are Key-Value Memories.** The FFNs layer in the transformer block typically includes weights $\mathbf{K} \in \mathbb{R}^{h \times d}$, $\mathbf{V} \in \mathbb{R}^{d \times h}$, where $h$ is embedding size and $d$ is the dimension of the hidden layer (usually $d = 4h$), and a non-linear activation function $\sigma(\cdot)$. For an input $\mathbf{x} \in \mathbb{R}^h$, the output $\mathbf{y} \in \mathbb{R}^h$ can be calculated as follows:

$$\mathbf{y} = \mathrm{FFN}(\mathbf{x}; \mathbf{K}, \mathbf{V}) = \sigma(\mathbf{x} \cdot \mathbf{K}) \cdot \mathbf{V}. \tag{1}$$

More precisely, for each column $\mathbf{K}_{:,i}$ and row $\mathbf{V}_{i,:}$, Equation 1 can be rewritten as:

$$\mathbf{y} = \sigma(\mathbf{x} \cdot \mathbf{K}) \cdot \mathbf{V} = \sum_{i=1}^{h} \sigma(\mathbf{x} \cdot \mathbf{K}_{:,i}) \cdot \mathbf{V}_{i,:} \tag{2}$$

We follow Lample et al. (2019); Geva et al. (2021; 2022) regarding each column in $\mathbf{K}$ as a key vector and each row in $\mathbf{V}$ as the value vector, the output of an FFNs network can be viewed as a weighted sum of value vectors based on the activation scores $\sigma(\mathbf{x} \cdot \mathbf{K})$. Viewing FFNs as collections of smaller units facilitates understanding of the structure within FFNs and the method 3.2 to decompose FFNs.

**Mixture-of-Experts** In transformers, MoEs is often implemented by replacing the original FFNs with a group of parallel FFNs and introducing a gating module. Supposing there are $N$ experts: $\{\mathrm{FFN}^n(\cdot; \mathbf{K}^n, \mathbf{V}^n) | n \in [1, N]\}$, the gating module $g(\cdot; \mathbf{G}, k)$, defined with its parameters $\mathbf{G}$ and an integer $k$, is to map input $\mathbf{x}$ to a score distribution of experts $g(\mathbf{x}; \mathbf{G}, k) \in \mathcal{R}^N$. Typically, $g$ is implemented with a simple linear layer followed by a $\mathrm{softmax}$ function and a $\mathrm{Top}\text{-}k$ function. Given an input $\mathbf{x} \in \mathbb{R}^h$, let $\mathbf{y}_n = \mathrm{FFN}^n(\mathbf{x}; \mathbf{K}^n, \mathbf{V}^n)$ be the output of the $n$-th expert, and then the output $\mathbf{y} \in \mathbb{R}^h$ of can also be summarized as the weighted sum of the output from all experts:

$$\mathbf{y} = \sum_{n \in N} g_n(\mathbf{x}; \mathbf{G}, k) \, \mathrm{FFN}^n(\mathbf{x}; \mathbf{K}^n, \mathbf{V}^n) = \sum_{n \in N} g_n(\mathbf{x}; \mathbf{G}, k) y_n, \tag{3}$$

When $k$ for $\mathrm{Top}\text{-}K$ is smaller than $N$, only a subgroup of experts is involved in the computation.

### 3.2 EMERGENT MIXTURE-OF-EXPERT

As our research goals mainly focus on how EM influences fone-tuning stage, a preferred approach to externalize EM into explicit MoEs models should not introduce additional parameters, training, and data, which may result in impractical or undesired biases. In this work, we adopt a method in ablation study of the MoEfication paper Zhang et al. (2022b). As illustrated in Figure 2, the method contains two steps: (1) clustering-based expert construction and (2) avg-k gating. It is simple and fulfills our needs, as demonstrated in our experiments.

**Clustering-based Experts Construction.** Since key-value pairs with similar key vectors tend to be co-activated, we split them into separate experts according to their key vectors. Specifically, given a trained FFNs layer $\mathrm{FFN}(\cdot; \mathbf{K}, \mathbf{V})$, we perform constrained clustering (Malinen & Fränti, 2014) to partition all key vectors $\mathbf{K}$ into $N$ experts on average, so each group contains $\frac{d}{N}$ key-value pairs. Denoting the indices of keys in the $i$-th group as $E_i \subset [d]$, for $\forall j \in E_i$, we extract key-value pair $(\mathbf{K}_{:,j}, \mathbf{V}_{j,:})$ to form the $i$-th expert $\mathrm{FFN}(\cdot; \mathbf{K}^i, \mathbf{V}^i)$, as depicted in Figure 2(b). After that, the computation of each expert proceeds as Equation 1.

**Avg-k Gating.** Given an input $\mathbf{x}$ and $N$ experts, a qualified gate for EMoE should route the input $\mathbf{x}$ to the experts who contribute most to the model's output. We construct the gating module by averaging

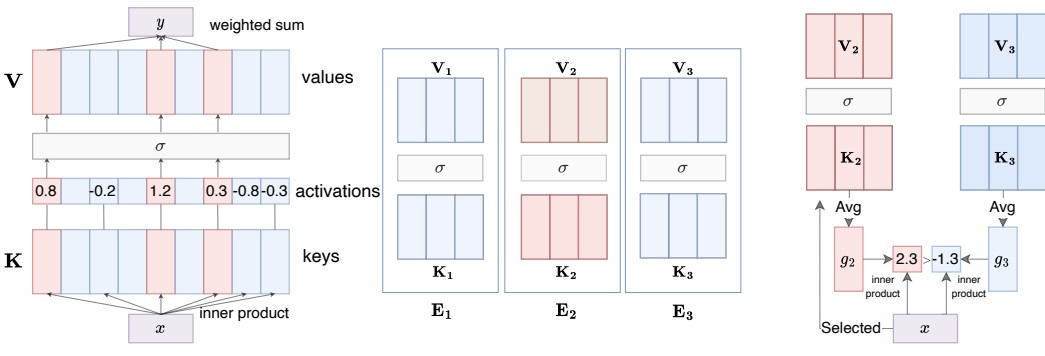

(a) spares and implicit modular FFN   (b) explicit experts via keys clustering   (c) avg-k gating

Figure 2: (a) Existing literatures (Geva et al., 2021; 2022) suggest that the FFNs in transformers can be viewed as key-value memories. They regarded the input as a query, the first layer as keys, and the second as values. Given an input, keys are sparsely activated (marked in red). Most of the values don't impact the output. (2) The FFNs block can be partitioned into experts by clustering keys. (3) Afterward, experts' keys averages are used as the gating weights. The inner product between **x** and gating weights are used to select experts.

each expert's keys. The gating function is usually implemented by a single layer $\mathbf{G} \in \mathcal{R}^{h \times N}$, in the avg-k gating's case, the weights in $i$-th column $\mathbf{G}_{:,i}$ can be calculated as follows:

$$\mathbf{G}_{:,i} = \mathrm{Avg}(\mathbf{K}^i, \mathrm{dim=0}). \tag{4}$$

And then the corresponding gating score for $i$-th expert is:

$$g_i(\mathbf{x}; \mathbf{G}, k) = \begin{cases} 1 & \text{if } i \in \mathrm{Top\text{-}K}(\mathbf{x} \cdot \mathbf{G}; k, \mathrm{dim=1}) \\ 0 & \text{else} \end{cases} \tag{5}$$

where $\mathrm{Top\text{-}K}(\cdot; k, \mathrm{dim})$ return indices of $k$ largest element of the given input along a given dimension. Using 0,1 score, avg-k gating reduces the weights' influence in Equation 3). Notably, observe:

$$\mathbf{x} \cdot \mathbf{G}_{:,i} = \mathbf{x} \cdot \mathrm{Avg}(\mathbf{K}^i, \mathrm{dim} = 0) = \mathbf{x} \cdot \frac{N}{d} \sum_{j \in E_i} \mathbf{K}^i_{:,j} = \frac{N}{d} \sum_j a_j. \tag{6}$$

A larger value of gating score $g_i$ implies more activated keys within the corresponding expert. Consequently, the expert could potentially contribute more to the output **y** for input **x**. During downstream tuning, gating weights are updated along with the FFNs parameters using Equation 4.

## 3.3 DISCUSSION

Though adopting a method from MoEfication (Zhang et al., 2022b), this work has a distinct motivation, leading to different model configurations and testing scenarios. MoEfication is motivated by the observation that FFNs layers in pre-trained transformers are sparsely activated (many neurons are unused for inputs). It splits FFNs into a sparse MoE, aiming to reduce the computational cost. MoEfication focuses on precise expert construction and gating to preserve performances while improving inference efficiency. In sharp contrast, EMoE wants to externalize the emergent modular nature of the pre-trained transformers so that the experts are sparsely updated and further encouraged to achieve distinct specializations. So, we decompose FFN into sparse MoE before fine-tuning on downstream tasks. Though similar methods are applied, EMoE provides a different view about what we can benefit from emergent modularity, and this is what we want to share with the community with this work. As one of our most related works, GMoE (Li et al., 2023) introduces MoEs by copying the trained FFNs. They further demonstrate that introducing MoEs in every two layers leads to significant degradation. Accordingly, we only introduce EMoEs in certain layers since efficiency isn't our primary concern. We also conduct ablation studies about MoEfying every two layers and get consistent results; more details are in Appendix A.5.2 Table 19.

## 4 EXPERIMENTS

**Experimental Configurations** We evaluate EMoE on vision and language tasks. All experiments are repeated 3 times independently. We present the average in the main section, while full results are in Appendix A.5.1. Detail settings can be found in each experiment part. **Tasks and models**: In vision tasks, we employ ViT-Small (22M) and ViT-Base (86M) on 4 datasets from the Domainbed (Gulrajani & Lopez-Paz, 2021) for benchmarking vision OOD performance. In language

tasks, we employ GLUE (Wang et al., 2019b) and GLUE-X (Yang et al., 2023) for benchmarking language ID and OOD performance. We evaluate EMoE with a wide range of pre-trained language models, including BERT-Base/Large (110M/340M), and GPT2-Small/XL (124M/1.5B). For more details about datasets and evaluation metrics, please refer to the Appendix A.3. **Hyper-parameters**: those unrelated to MoEs, like the learning rate, remain consistent with the baselines. For others, guided by Li et al. (2023), we explore MoEs layers in {last two even layers, last one even layer}. Comparable hyper-parameter searches are conducted for both GMoE and EMoE for the number of experts $N$ and top-k. In vision tasks, as highlighted by (Gulrajani & Lopez-Paz, 2021), the hyper-parameter search has a profound impact on outcomes. Consequently, we search with a relatively large scope: for GMoE, $N$ is searched within {4, 6, 8}, and top-k within {2, 3, 4}; for EMoE, $N$ is sought within {6, 12, 24}, and top-k within {2, 4, 8}. For Languag tasks, GMoE explores $N$ within {4, 8} and top-k within {1, 2}. For EMoE, $N$ is fixed at 64, with top-k explored within {16, 32}. Our ablation study indicates that while hyper-parameter search yields superior performance, adhering to a top-k/$N = 0.5$ for EMoE consistently brings improvement over dense counterparts.

**Baselines** On Domainbed, our baselines include (1) vanilla ViT, as it is a strong OOD baseline under fair configurations suggested by Gulrajani & Lopez-Paz (2021). (2) GMoE (Li et al., 2023), which is the latest state-of-the-art. GMoE constructs MoEs by replicating FFNs from the two-to-last and fourth-to-last transformer blocks of a pre-trained ViT and therefore is the most relevant to EMoE. For language tasks, besides vanilla backbone and GMoE, we implement (3) noisy tuning (Wu et al., 2022), which also improves adaptation for free by adding uniform distribution noise proportional to the standard deviation of the pre-trained weights before tuning. (4) EMoE-learn, an ablation method, where the gating function is learned (same as **GMoE**) during fine-tuning instead of employing avg-k gating. This helps us better understand the effect of avg-k gating.

## 4.1 FULL FINE-TUNING PERFORMANCE

Table 1: Overall OOD performances with 3 selection criteria. All the reported results are obtained following the Domainbed codebase. The best result is highlighted in **bold**. In cases where results are the same, the best result is determined by the smallest standard deviation. EMoE demonstrates comparable results to GMoE.

| Results with ViT-small (22M) backbone | | | | | | Results with ViT-base (86M) backbone | | | | | |
|---|---|---|---|---|---|---|---|---|---|---|---|
| **Algorithm** | **PACS** | **VLCS** | **Office** | **Terra** | **Avg** | **Algorithm** | **PACS** | **VLCS** | **Office** | **Terra** | **Avg** |
| Train-validation selection criterion | | | | | | Train-validation selection criterion | | | | | |
| ViT | 86.9 | **79.7** | 73.0 | 44.0 | 70.90 | ViT | 89.1 | **80.7** | 77.2 | 47.3 | 73.58 |
| GMoE | 87.7 | 79.6 | 73.1 | 45.4 | 71.45 | GMoE | **90.0** | 80.4 | 77.0 | **49.2** | **74.15** |
| EMoE-learn | 87.2 | 79.6 | 72.5 | **46.1** | 71.35 | EMoE-learn | 89.8 | 80.6 | 76.5 | 48.7 | 73.90 |
| EMoE | **87.8** | 79.5 | **73.1** | 45.9 | **71.58** | EMoE | 89.4 | 80.7 | **77.3** | 48.5 | 73.98 |
| Leave-one-domain-out selection criterion | | | | | | Leave-one-domain-out selection criterion | | | | | |
| ViT | 86.1 | 79.7 | **73.3** | 45.0 | 71.03 | ViT | 88.9 | 80.8 | **77.5** | 46.1 | 73.33 |
| GMoE | 86.5 | 80.5 | 73.1 | 45.3 | 71.35 | GMoE | 89.3 | 81.0 | 76.7 | 50.1 | 74.28 |
| EMoE-learn | **86.8** | 79.6 | 72.6 | 45.8 | 71.20 | EMoE-learn | 89.3 | 81.2 | 76.5 | **50.5** | 74.38 |
| EMoE | 86.8 | **80.6** | 73.3 | **46.1** | **71.70** | EMoE | **89.6** | **81.6** | 77.4 | 50.0 | **74.65** |
| Test-domain selection criterion | | | | | | Test-domain selection criterion | | | | | |
| ViT | 86.5 | 78.2 | 73.1 | 44.0 | 70.45 | ViT | 88.8 | 79.0 | 77.2 | 46.7 | 72.93 |
| GMoE | 87.2 | 79.0 | **73.4** | 45.3 | 71.23 | GMoE | 89.7 | 79.0 | 77.0 | **48.8** | 73.63 |
| EMoE-learn | 87.4 | **79.1** | 72.8 | 45.4 | 71.18 | EMoE-learn | **89.7** | **79.7** | 76.6 | 48.7 | 73.68 |
| EMoE | **87.6** | 79.0 | 73.3 | **45.5** | **71.35** | EMoE | 89.7 | 79.7 | **77.5** | 48.8 | **73.93** |

**We test the EMoE's OOD performance on Domainbed.** Domainbed provides comprehensive vision OOD evaluations (one result is aggregated with 30 experiments), and the outcomes vary marginally. Moreover, Gulrajani & Lopez-Paz (2021) indicates that vanilla full fine-tuning with fair hyper-parameter search is a strong baseline compared with specifically designed methods like Invariant Risk Minimization (Arjovsky et al., 2019). More dataset details are in appendix A.3.1. According to Table 1: (1) Overall, EMoE outperforms ViT and GMoE (except ViT-base Train-validation setting, upper right). (2) Compared with EMoE, EMoE-learn incorporates a learned gate. While it surpasses avg-k gating in certain scenarios (ViT-small Terra Train-validation), it can also lead to a performance drop compared with vanilla ViT. Its overall performance is lower than EMoE. (3) In tasks where the dense model is strong (like Office), EMoE performs better than other MoEs methods. One possible reason is that the avg-k gating reduces the influence of gating weights ($g_n(\mathbf{x}; \mathbf{G}; k)$ in Equation 3), making it more like the dense model in such scenarios.

Table 2: ID performance on GLUE tasks with different backbones and algorithms. All the reported results are obtained from 3 independent experiments. The average accuracy (Avg) is reported along with the relative improvement compared to the baseline. The best result is highlighted in **bold**.

| Backbone | Algorithm | MRPC | CoLA | RTE | STSB | SST2 | Avg |
|---|---|---|---|---|---|---|---|
| BERT Base | baseline | 88.45 | 60.67 | 68.95 | 87.87 | 91.97 | 79.582 |
| | noisy tuning | 88.43 | 61.79 | 71.36 | 88.27 | 92.32 | 80.43(+0.85) |
| | GMoE | 88.63 | 61.25 | 70.28 | 88.63 | 92.28 | 80.21(+0.63) |
| | EMoE-learn | 89.05 | **62.46** | **70.40** | 88.47 | 92.58 | **80.59(+1.01)** |
| | EMoE | **89.45** | 61.55 | 69.68 | **88.71** | **92.89** | 80.46(+0.87) |
| BERT Large | baseline | 89.82 | 65.41 | 74.89 | 89.87 | 93.50 | 82.70 |
| | noisy tuning | 90.42 | 64.75 | 73.41 | 90.05 | 93.65 | 82.46(-0.24) |
| | GMoE | **91.24** | 64.90 | 74.24 | 90.00 | 93.58 | 82.79(+0.09) |
| | EMoE-learn | 90.57 | 65.51 | 74.72 | 90.22 | **93.73** | 82.95(+0.25) |
| | EMoE | 90.74 | **65.79** | **76.17** | 90.31 | 93.58 | **83.32(+0.62)** |
| GPT2 Small | baseline | 84.46 | 47.07 | 67.15 | 86.29 | 92.13 | 75.42 |
| | noisy tuning | 84.15 | 46.16 | 67.51 | 86.09 | 92.13 | 75.21(-0.21) |
| | GMoE | 85.07 | 47.77 | 67.51 | 86.57 | 92.35 | 75.85(+0.43) |
| | EMoE-learn | **85.73** | 47.24 | 67.99 | **86.66** | 92.35 | 75.99(+0.57) |
| | EMoE | 85.40 | **48.00** | **68.95** | 86.64 | **92.70** | **76.34(+0.92)** |

**We evaluate EMoE's ID performance on 5 GLUE tasks**. According to Table 2: (1) On average, EMoE and EMoE-learn outperform other baselines. (2) Among the two methods that do not introduce additional parameters, EMoE significantly outperforms noisy tuning. (3) EMoE provides stable improvements over baselines across different settings, demonstrating its generality.

## 4.2 LoRA-TUNING PERFORMANCE

Table 3: ID and OOD performance of EMoE and baseline models. All the reported results are obtained from 3 independent experiments. OOD Metrics (averaged over 14 OOD tasks, lower is better) provide additional information for out-of-distribution generalization. The best result is highlighted in **bold**.

| Algorithm | MRPC | CoLA | RTE | STSB | SST2 | QNLI | QQP | MNLI | ID-Avg | OOD |
|---|---|---|---|---|---|---|---|---|---|---|
| | | | | BERT-Large (340M Parameters) | | | | | | |
| LoRA | 89.97 | 63.40 | 72.92 | **90.51** | 93.16 | 92.20 | 87.21 | 85.40 | 84.35 | 4.86 |
| Block | 89.34 | 62.10 | 71.96 | 90.39 | 93.35 | 92.04 | 88.45 | 86.20 | 84.23(-0.12) | 4.95 |
| GMoE | 89.45 | 63.80 | 72.56 | 90.29 | **93.85** | 92.32 | 87.99 | 85.92 | 84.52(+0.18) | **4.04** |
| EMoE-block | 89.77 | 63.25 | 71.60 | 90.31 | 93.69 | 92.09 | 88.08 | **86.21** | 84.38(+0.03) | 5.89 |
| EMoE-learn | 89.87 | 64.00 | 71.36 | 90.48 | 93.65 | **92.40** | 87.55 | 85.62 | 84.37(+0.02) | 4.66 |
| EMoE | **90.85** | **65.33** | **75.21** | 90.43 | 93.50 | 92.23 | 87.74 | 85.43 | **85.09(+0.74)** | 4.37 |
| | | | | GPT2-XL (1.5B Parameters) | | | | | | |
| LoRA | 86.83 | 60.88 | 78.70 | 89.07 | 95.18 | 91.84 | 87.41 | 86.93 | 84.61 | 5.61 |
| Block | 86.59 | 61.18 | 79.78 | 89.08 | 95.45 | 91.88 | 87.71 | 86.95 | 84.83(+0.22) | 5.13 |
| GMoE | 87.02 | **62.81** | 79.78 | 89.21 | 95.41 | 92.18 | 89.10 | **87.17** | 85.34(+0.73) | 4.33 |
| EMoE-block | 87.86 | 62.88 | **80.05** | 89.18 | **95.49** | 92.10 | 89.69 | 86.87 | **85.52(+0.91)** | 5.71 |
| EMoE-learn | **87.93** | 61.50 | 79.90 | **89.48** | 95.18 | 92.33 | **89.71** | 87.00 | 85.38(+0.77) | 4.40 |
| EMoE | 87.75 | 62.27 | 80.02 | 89.37 | 95.41 | 92.10 | 89.58 | 87.06 | 85.45(+0.84) | **3.88** |

With the increasing scale of pre-trained models, parameter-efficient tuning (Houlsby et al., 2019) gets popular. Observing weaker results with LoRA-tuned ViT-Large on Domainbed compared to ViT-Base, we omit LoRA tuning on Domainbed. GLUE-X (Yang et al., 2023) is a recently introduced OOD dataset for the GLUE benchmark. Therefore, we assess EMoE's ID performance on GLUE and OOD performance on GLUE-X with the standard LoRA tuning. Notably, LoRA weights are only added in attention, with all other weights frozen. Therefore, the only difference between EMoE and its dense counterpart is FFNs activation. For the *OOD metric*, we follow GLUE-X (Yang et al., 2023) and employ the Friedman rank (Friedman, 1940) $\text{rank}_f = \frac{1}{n}\sum_{i=1}^{n}\text{rank}_i$. For each method under the same backbone, $\text{rank}_i$ is produced based on the best result and the average result for each dataset. For the 13 OOD tasks used in GLUE-X, each method generates 26 $\text{rank}_i$ values. The OOD results presented in Table 3 represent the mean of all these $\text{rank}_i$ values. The original results of each task can be found in the Appendix A.5.1. For *algorithm configurations*, since GMoE copies the FFNs of the pre-trained model to form the MoEs, it is meaningless if the MoEs parameters aren't tuned. Thus, in Table 3, we conduct experiments with LoRA tuning for GMoE and also tune the transformer block where the original FFNs is replaced. For comparison, we also (1) LoRA-tune dense model and fine-tune the transformer block at the corresponding layer (denoted "Block"); (2) LoRA-tune EMoE with the transformed block also updated (denoted "EMoE-block").

According to Table 3, EMoE continues to enhance the downstream performance in LoRA tuning: (1) EMoE demonstrates significant enhancements compared to LoRA and doesn't introduce additional trainable parameters or procedures. Notably, EMoE achieves comparable results with GMoE on BERT-large and outperforms it on GPT2-XL. (2) When the blocks are tuned, EMoE also brings improvements (EMoE-block vs. Block) (3) Consistent with the full-finetuning findings, EMoE exhibits higher stability than EMoE-learn and delivers superior overall results. (4) Besides vision, MoEs also improve OOD performance in language tasks (GMoE vs. Block, EMoE vs. LoRA).

## 5 ANALYSIS

### 5.1 HOW DOES EMoE IMPROVE FINE-TUNING PERFORMANCE?

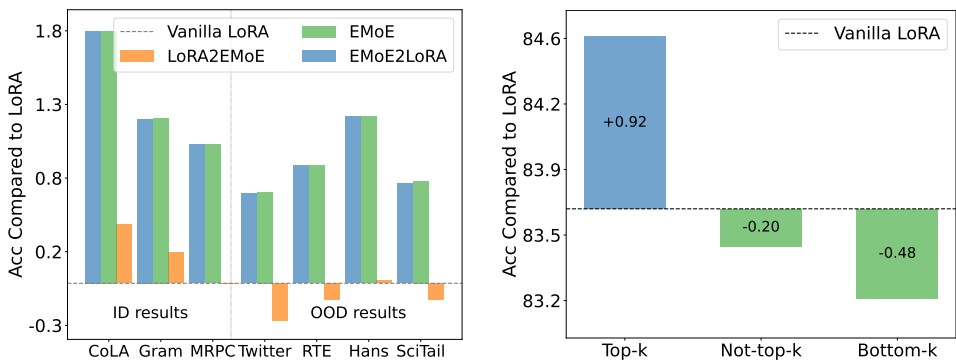

Figure 3: Left: ID and OOD results compared with LoRA for validating EMoE's training & inference effects. Right: sparse activated training results with different expert selections.

EMoE achieves notable improvements in both ID and OOD scenarios. However, it is non-trivial that simply transforming the pre-trained FFNs into MoEs before fine-tuning can yield such benefits (especially in LoRA tuning where the MoEs part is not updated). Therefore, we investigate the reasons behind the enhanced fine-tuning performance. *To decouple the impact of updating expert parameters, we focus on LoRA tuning.* We use BERT-Large as the backbone model.

**EMoE benefits LoRA weights learning instead of influencing inference.** EMoE and dense model only differ in FFNs activations. Such difference might (1) directly impact outputs during testing and (2) influence the parameter updating during training. In light of this, we propose two variants and compare them with vanilla LoRA tuning and LoRA tuning with EMoE: (a) LoRA2EMoE: Using LoRA to fine-tune the original model and split it into its MoEs counterpart. If the test results better the vanilla LoRA-tuning, we can infer that the sparse activation has an impact on model inference. (b) EMoE2LoRA: Using LoRA to fine-tune an EMoE model and merge experts back into their original FFNs netowrk. If no changes occur, it implies that the primary reason why EMoE brings better generalization is that it affect the parameter updating of fine-tuning stage. According to Figure 3 left, doing sparse activation during testing does not contribute to better generalization. (LoRA2EMoE). Furthermore, when merging LoRA-tuned EMoE blocks back into the original FFNs, the performance remains significantly better than the vanilla LoRA tuning (EMoE2LoRA vs. LoRA) and is almost identical to EMoE. Please refer to Appendix A.5 Table 16 for full results.

**EMoE masks neurons with negative transfer impacts.** The only difference between EMoE and vanilla LoRA tuning is EMoE blocks some activated neurons during training by Top-k expert selection. Based on these, we hypothesize that EMoE's effects stem from preventing negative knowledge transfer from blocked neurons. Therefore, we investigate whether there are such negative transfers. Specifically, we study the following expert selection variants: (1) Bottom-k: select $k = 16$ experts who get the lowest scores; (2) Not-top-k: select experts who are not among the top-k experts. These variants are evaluated across 6 tasks from GLUE. The averaged outcomes are in Figure 3 Right. Full results can be found in Appendix A.5 Table 17. LoRA tuning results with "Bottom-k" and "Not-top-k" expert selections are worse than vanilla LoRA tuning. One may attribute this drop to the reduced number of employed neurons. To further assess this, we compare "Top-k" with "Bottom-k", where an equal number of neurons are used. Notably, "Bottom-k" significantly lags behind "Top-k". This further corroborates that masked neurons have negative transfer effects.

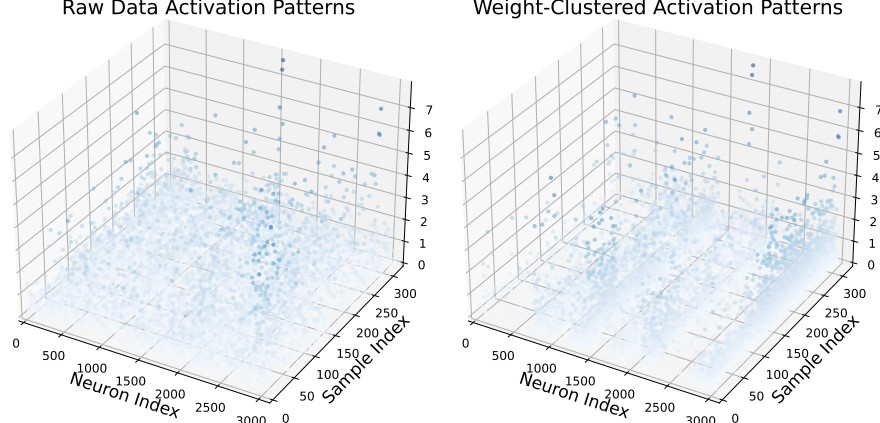

Figure 4: Left: Activations of neurons in FFNs a pre-trained ViT (Dosovitskiy et al., 2021). Right: By clustering the keys in the FFNs and rearranging the action values accordingly, modular activation patterns of neurons emerge. These co-activated patterns correspond to the modular structure within pre-trained FFNs. With appropriate partitioning, we can transform dense networks into modular networks.

## 5.2 ABLATION STUDIES

**Expert Constructing Methods** Figure 4 demonstrates that clustering can decompose modular components within the dense model. To provide further evidence that the EMoE's improvements stem from leveraging modular features rather than just sparse activation, we compare the results of (1) *clustering-based* expert construction and (2) *random* construction within the same setting of Section 5.1. The relative changes in averaged outcomes compared to the dense baseline are shown in

Table 4: Clustering-based and random expert constructing results

|  | Top-k | Bottom-K |
|---|---|---|
| Cluster | +0.92 | -0.48 |
| Random | -0.11 | -0.34 |

Table 4, while full results can be found in the appendix A.5, Table 17. It's noteworthy that while cluster top-k exhibits a significant improvement over dense, random top-k is conversely worse than dense baseline. This suggests that random construction can negatively impact gating. Moreover, when selecting weights with negative transfer under bottom-k selections, it's observed that cluster bottom-k also achieves lower results. In summary, clustering-based methods can externalize implicit modularity within the dense model. Within suitable frameworks like MoEs, such modularity can facilitate downstream tuning.

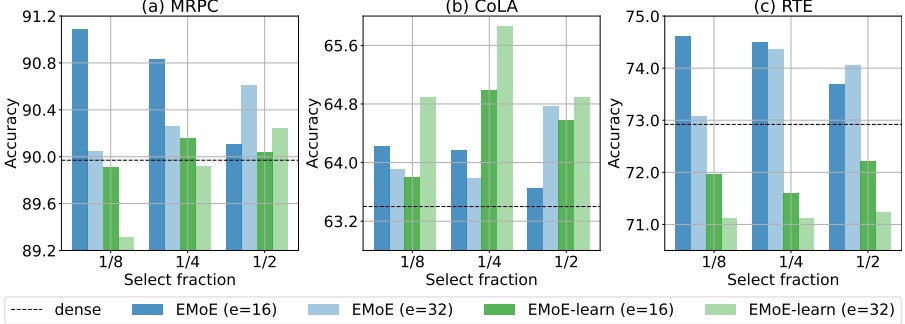

Figure 5: Results of 3 tasks for different experts splittings and expert select fractions (top-k / number of experts). 'e=16' and 'e=32' mean splitting FFNs into 16 and 32 experts, respectively.

**Expert Constructing Configurations** Beyond the settings detailed in the main results section, which are based on $N = 64$ experts and top-k $\in \{16, 32, 48\}$, we also present specific scenarios where $N \in \{16, 32\}$, and top-k varies within $\{2, 4, 8, 16\}$ in Figure 5. Notably, in each of these settings, (1) EMoE consistently surpasses the dense model, illustrating its insensitivity to hyper-parameters. (2) On average, avg-k gating exhibits superior performance than learned gating. Though learned gating (EMoE-learn) outperforms avg-k gating in a few specific settings (Figure 5 (b) and (e)). This is consistent with the earlier results in Section 4. Regarding how many EMoE layers should be introduced, our findings align with those discovered in GMoE, indicating that only a limited number of layers can be converted into the EMoE layer. If excessive EMoE layers are introduced, performance deteriorates. Taking GPT2-XL (48 layers) as an example, when introducing EMoE every 2 layers in the latter half, the performance averaged across 5 GLUE tasks

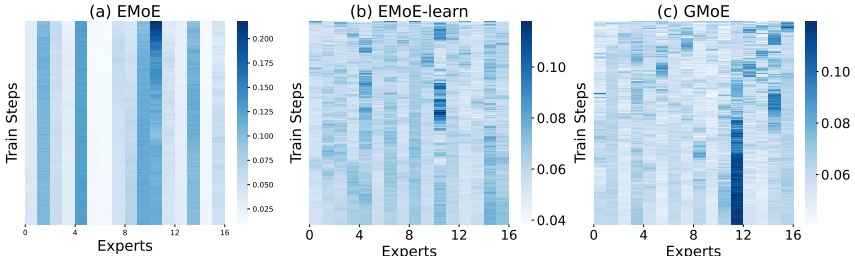

Figure 6: Expert selections during training with distinct gating functions (avg-k vs. learned) and expert types (modules from dense vs. repetitions of dense). The vertical axis illustrates training steps (top-down arrangement signifies begin-end); the horizontal axis represents expert selection frequency within 1K steps (deeper color implies a higher frequency). (a), (b) and (c) correspond to EMoE, EMoE-learn, and GMoE.

(79.36) matches that of the dense model (78.87). But when adopting EMoR every 2-layer for the entire model, the performance lags slightly behind that of the dense model (78.17) but surpasses EMoE-learn (75.87). For additional configurations, please refer to the appendix A.5.2 Table 19.

**How Expert Selection Changes During Training** To further understand avg-k gating and learned gating, we visualize expert selections of GPT2-XL during fine-tuning on 6 tasks with 16 experts. In Figure 6, we showcase the results for the largest dataset QNLI among them. Full results are available in the appendix A.6. Our observations are: (1) Both avg-k gating and learned gating converge, as indicated by the lower halves of the plots. (2) avg-k gating is more stable than learned gating (Figure 6 a vs. b). This might mitigate data inefficiency from gating inconsistencies across different stages of training (Zuo et al., 2022). (3) EMoE, with its heterogeneous experts, exhibits better load balancing than GMoE (Figure 6 b vs. c). In GMoE, all experts share identical initialization, whereas in EMoE, the experts are derived from FFNs with implicit modularity. This also suggests a good initialization can facilitate MoEs learning similar to EvoMoE (Nie et al., 2021).

**Performance Across Different Training set Volumes.** Previous research has indicated that modular architectures offer improved data efficiency (Bengio et al., 2020). Therefore, we conducted experiments with GPT2-XL on six tasks using varying proportions of original training data, and the results for all tasks are presented in Figure 7. It can be observed that EMoE consistently outperforms the dense across different data factions. EMoE achieves superior results even when using less than 20% of the data. On SST2, only using 50% data, EMoE shows comparable performance to the dense. More details can be found in the Appendix A.5.2 Table 20. This further underscores the benefits of incorporating modular structures.

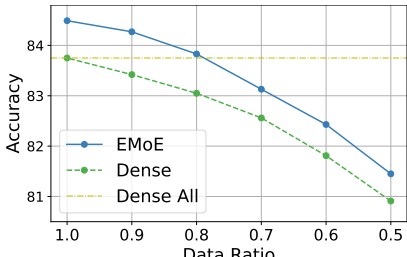

Figure 7: Average performance of EMoE with different proportions of training data.

## 6 CONCLUSION, LIMITATIONS AND FUTURE WORKS

**Conclusion**: In this work, we validate that exploiting the emergent modular structures in dense pretrained transformers improves downstream task ID and OOD performances. One possible reason is the modular structure can alleviate negative transfer effects presented in the pre-trained model. We hope our findings could deepen the understanding of neural networks' modularity, further helping the community develop more sophisticated modular neural architectures and utilizing existing pre-trained models. **Limitations**: Our primary objective was to investigate the utility of EM, and thus, we predominantly adopted the techniques from MoEfication for decomposition. We encourage further research to propose improved algorithms for harnessing EM. Our research findings have not been validated on more challenging tasks (e.g., Mathematical Reasoning (Imani et al., 2023)). While our analysis was primarily conducted on models with a maximum parameter count of 1.5B, we validate the scaling up ability of EMoE to Llama-30B. **Future works**: Subsequent research could delve into better methods for modular decomposition and investigate the modularity in foundation models. Benefiting from EMoE's ability to avoid negative transfer, future researchers can also exploit the emergent modularity in settings such as multi-task learning and continual learning. Moreover, it is observed that modularity structures emergent and remain stable after approximately one-fourth of the pre-training phase (Li et al., 2022; Zhang et al., 2023). Future investigations could investigate leveraging the modularity of dense models during the pre-training process.

## 7 REPRODUCIBILITY

In the Experiment Configuration section 4, we first introduce the additional configurations introduced by EMoE relative to the Dense Model, as well as the baseline settings. Subsequently, we provide high-level experiment settings for each experimental group (Domainbed at 4.1, full fine-tuning GLUE tasks at section 4.1, LoRA tuning at 4.2). In Appendix A.3, we provide detailed information on the datasets used, the codebase, and the evaluation metrics. We also outline more general configurations, such as learning rates and batch sizes for the respective tasks. Beyond the main paper, we include the original code for the experiments and log files for certain experimental results in the supplementary materials. Each section of code includes a README.md file that explains the experimental settings required to replicate the results.

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

## A APPENDIX

### A.1 ADDITIONAL EXPERIMENT RESULTS

#### A.1.1 MULT-TASKS SETTING

Table 5: T5-base ID performances. All tasks are from SuperGLUE so we omit the prefix "Super-GLUE" for each tasks.

| Experts (N) | topk (k) | boolq | cb | wic | wsc.fixed | rte | copa | test avg |
|---|---|---|---|---|---|---|---|---|
| Baseline | | 82.14 | 85.71 | 65.83 | 34.62 | 74.10 | 52.00 | 65.73 |
| 8 | 2 | 80.67 | 89.29 | 65.52 | 36.54 | 79.86 | 56.00 | 67.98 |
| 16 | 4 | 81.16 | 89.29 | 68.34 | 51.92 | 74.10 | 44.00 | 68.14 |
| 32 | 8 | 80.12 | 78.57 | 70.85 | 63.46 | 82.73 | 64.00 | **73.29** |
| 32 | 16 | 81.04 | 75.00 | 72.41 | 57.69 | 78.42 | 54.00 | 69.76 |

In our analysis 5.1, we have identified that the improvement brought by EMoE is likely associated with mitigating negative transfer. Inspired by this, we choose a multi-task learning setting where negative transfer might be more pronounced. We adopt the codebase[1] from ATTEMPT (Asai et al., 2022). For the in-domain (ID) scenario, we follow the settings outlined in ATTEMPT and select six tasks from the Super-GLUE benchmark (Wang et al., 2019a). For the out-of-domain (OOD) scenario, we take two larger natural language inference (NLI) datasets MNLI and QNLI, from GLUE

---

[1]https://github.com/AkariAsai/ATTEMPT

Table 6: T5-base OOD performances. 'SG' refers to 'SuperGLUE'.

| Experts (N) | topk (k) | mnli | qnli | wnli | rte | SG-rte | SG-cb | OOD avg |
|:---:|:---:|:---:|:---:|:---:|:---:|:---:|:---:|:---:|
| Baseline | | 86.2 | 92.42 | 50.00 | 61.87 | 62.59 | 32.14 | 51.65 |
| 8 | 2 | 86.27 | 92.18 | 50.00 | 64.03 | 62.59 | 32.14 | 52.19 |
| 16 | 2 | 86.44 | 92.59 | 52.78 | 64.03 | 56.83 | 39.29 | **53.23** |
| 32 | 8 | 86.56 | 92.49 | 58.33 | 64.03 | 61.87 | 28.57 | 53.20 |

as our ID training data. We subsequently conducted direct testing on four additional NLI datasets from different domains. All hyperparameters unrelated to MoEs are kept consistent with the baseline, and we have listed the MoEs-related hyperparameters in the result table. Our observations are as follows:

1. EMoE exhibits a substantial improvement compared to the baseline. In the in-domain (ID) setting, the highest improvement reached 7.56, even considering the average performance across the six tasks. In the out-of-domain (OOD) setting, the highest average OOD result across the four datasets improved by 1.58.

2. Across various settings of N and K, EMoE consistently outperforms the vanilla fone-tuning. Within the hyperparameter search space specified in our paper, EMoE consistently improves at least 2 points over the baseline in the in-domain (ID) setting. This emphasizes the effectiveness of EMoE and EMoE's robustness to the explored hyperparameter range.

### A.1.2 INSTRUCTION-TUNING LLMs SETTING

Table 7: Instruction full-tuned Llama2-7B's MMLU scores. Times are wall-clock computation times. The term 'times' in the subsequent tables refers to the same concept. 'w/o' refers to 'without'.

| Experts (N) | topk (k) | MMLU score | times (s) | FLOPS ($10^{16}$) |
|:---:|:---:|:---:|:---:|:---:|
| w/o tuning | - | 46.79 | - | - |
| full tuning | - | 46.5 | 4988 | 8.97 |
| 64 | 16 | **48.08** | 5036 | 9.12 |
| 64 | 32 | 47.44 | 5041 | 9.24 |

Table 8: Instruction LoRA-tuned Llama2-7B's MMLU scores.

| Experts (N) | topk (k) | MMLU score | times (s) | FLOPS ($10^{17}$) |
|:---:|:---:|:---:|:---:|:---:|
| w/o tuning | - | - | - | - |
| LoRA-tuning | - | 46.96 | 1396 | 6.92 |
| 64 | 16 | **47.58** | 1545 | 7.03 |
| 64 | 32 | 47.37 | 1521 | 7.13 |

To further prove that the main conclusion EMoE still holds for larger LLM, we use the Alpaca dataset to instruction-tune the Llama series models (Touvron et al., 2023) and evaluate it on the MMLU benchmark (Hendrycks et al., 2021). We have observed the following:

1. Across model sizes of 7B, 13B, and 30B, as well as settings such as full-finetuning and Lora tuning, EMoE consistently yields improvements relative to the baseline.

2. The choice of K and N proportions remains applicable even in larger-scale models. While variations may be in different settings, they consistently outperform the baseline. This suggests that although additional hyperparameters are introduced, they do not lead to usability challenges.

### A.2 COMPARISON BETWEEN EMoE, MOEFICATION AND GMoE

Having observed that FFN layers in pre-trained Transformers are sparsely activated (many neurons are unused for inputs), MoEfication splits transforms FFNs into a sparse MoE, aiming to approximate the functionality of the original FFNs to reduce the computational cost, further improving inference efficiency. Besides, the GMoE makes multiple replicates of the original FFN layer and introduces a learned gate to form a MoE architecture. They claim that such architecture could improve

Table 9: Instruction LoRA-tuned Llama-30B's MMLU scores.

| Experts (N) | topk (k) | MMLU score | times (s) | FLOPS ($10^{18}$) |
|---|---|---|---|---|
| w/o tuning | - | 51.5 | - | - |
| LoRA-tuning | - | 56.18 | 6943 | 2.25 |
| 256 | 128 | **57.11** | 6955 | 2.25 |
| 256 | 128 | 56.64 | 6974 | 2.24 |

Table 10: Comparison between EMoE, Moefication and GMoE

| Aspect | EMoE | Moefication | GMoE |
|---|---|---|---|
| Research Problem | Exploit the emergent modularity during fine-tuning pre-trained transformers. | Approximate FFNs with sparse MoE to improve inference efficiency. | Validate the OOD improvements brought by Sparse MoE architectures. |
| Methods | Split FFNs | Split FFNs | Copy of FFNs |
| Practicality | No additional trainable parameters. Experts can be recomposed into the dense model so that models can be deployed as a standard model. | May need re-training on the original task. May suffer from inference latency owing to the specific implementation of MoE architectures. | Additional trainable parameters are introduced. May suffer from inference latency owing to the specific implementation of MoE architecture. |
| Contributio | Significant general improvement without adding parameters and not depending on the specific implementation. | Improved inference efficiency (depending on the specific implementation of MoE), but performance drop. | Significant OOD improvement with additional parameters and specific implementation. |

OOD performance from their theoretical perspective of algorithmic alignment framework. MoEfication and GMoE do not touch on how emergent modularity influences the training stage. The table below illustrates a mdetailed comparison of these works. These differences are summarized in Table 10.

## A.3 DATASETS AND EVALUATION METRICS

### A.3.1 DOMAINBED

Table 11: Used dataset information from Domainbed

| Dataset | PACS | VLCS | OfficeHome | TerraInc |
|---|---|---|---|---|
| #Domains | 4 | 4 | 4 | 4 |
| Classes | 7 | 5 | 65 | 10 |
| Images | 9,991 | 10,729 | 15,588 | 24,788 |

The four datasets (PACS (Li et al., 2017), VLCS (Fang et al., 2013), Office-Home (Venkateswara et al., 2017), and Terra Incognita (Beery et al., 2018)) are selected from Domainbed. Each dataset comprises 4 distinct domains(PACS: {art, cartoons, photos, sketches}, VLCS: {Caltech101, LabelMe, SUN09, VOC2007}, Office-Home: {art,clipart, product, real}, and Terra Incognita: {L100, L38, L48, L46}). One or two domains' data are sequentially designated within a single training for OOD evaluation. For example, when training on PACS, {art, cartoons} could be selected as ID training data, while {photos, sketches} are designated for OOD testing. This configuration results in $C_4^2 + C_4^1 = 10$ distinct training processes within each dataset. Suppose there are $d_{tr}$ ID domains, "Train-validation" means selecting OOD test checkpoints based on ID accuracies from the validation subsets of all $d_{tr}$ ID domains; "Leave-one-domain-out" means leaving one selected ID domain as a validation set, doing training on $d_{tr} - 1$ domains; "Test-domain" means selection based on limited access to test domains and selection based on these results. The final results are aggregated with

selection criteria provided by Domainbed[2]. As a result, even a variation of 0.1 in the benchmark outcomes signifies a significant improvement.

In our experiments, all the hyper-parameters, like training steps, learning rates, and weight decay, except those related to MoEs, strictly follow GMoE.

### A.3.2 GLUE

Each task involves one to four OOD tasks from GLUE-X (Yang et al., 2023), resulting in 13 OOD tasks in total. To illustrate, consider the Sentiment Analysis task: we first fine-tune models on SST-2 from GLUE and report the validation results as ID performance, then use the test data from IMDB (Maas et al., 2011), Yelp (Zhang et al., 2015), Amazon (Kaushik et al., 2020) and Flipkart (Vaghani & Thummar, 2023) from GLUE-X for OOD testing.

In the *full fine-tuning*, to ensure convergence and reduce randomness, we train all models 10 epochs across 3 random seeds on each task. Each experiment does a hyper-parameter search on learning rates on [2e-5, 3e-5, 5e-5] as suggested by BERT (Devlin et al., 2019). The training batch size is 32. In the *LoRA tuning*, following LoRA (Hu et al., 2022) that tunes models with more epochs and larger learning rates than standard full fine-tuning, all models are trained 20 epochs on small and medium datasets and 5 epochs on large ones (like QNLI, MNLI, QQP). The learning rate is searched in [2e-4, 3e-4, 5e-4]. All methods are implemented with LoRA_rank=8 and LoRA_alpha=16. The training batch size is 16 due to a larger model size. Other settings like max_lengt following the codebase from hugging face[3]. After training on GLUE, we directly test the selected models on GLUE-X with the data from the official repo[4].

Table 12: Language tasks and corresponding ID and OOD datasets.

| Task | ID-dataset | size | OOD-dataset | size |
|---|---|---|---|---|
| Paraphrase | MRPC | 4,076 | Twitter | 16,777 |
| | QQP | 404,301 | Twitter | 16,777 |
| | | | MRPC | 4,076 |
| Linguistic Acceptability | CoLA | 9,594 | Grammar Test | 304,277 |
| Textual Entailment | RTE | 2,768 | SciTail | 26,527 |
| | | | HANs | 60,000 |
| Textual Similarity | STSB | 7,128 | SICK | 9,840 |
| Sentiment Analysis | SST2 | 68,223 | IMDB | 50,000 |
| | | | Yelp | 598,000 |
| | | | Amazon | 4,000,000 |
| | | | Flipkart | 205,041 |
| Question Answering NLI | QNLI | 110,206 | NewsQA | 119,525 |
| Natural Language Inference | MNLI | 412,313 | SICK | 9,840 |

### A.4 COMPUTATION COST AND MEMORY USAGE

#### A.4.1 EXPERIMENTS WITH PUBLIC MOES LIBRARY

Theoretically, EMoE does not introduce additional parameters compared to its dense counterpart. Although it adds computation in the gating portion within the MoEs layer, it omits a substantial amount of computation within the FFNs layer. For instance, the computation in the gating portion is of the order of $h \times N$, where h represents the model's hidden size, and N is the number of experts. In contrast, the complete computation in the FFNs layer is of the order of $(h \times h \times 4h) \times 2$, and sparse activations can reduce more than a quarter of this computation. Since $N \ll h$, theoretically,

---

[2]https://github.com/facebookresearch/DomainBed

[3]https://github.com/huggingface/transformers/tree/main/examples/pytorch/text-classification

[4]https://github.com/YangLinyi/GLUE-X

using EMoE within a single block should accelerate the forward pass of the model. However, in real deployment, we have observed that the hardware implementation of MoEs can result in EMoE being, on average, approximately 10% slower than the dense model. Additionally, the memory usage is also slightly higher, by less than 5%, compared to the dense model (16295MB v.s. 15865MB on CoLA, LoRA tuning GPT2-XL).

Each individual experiment was conducted on a single NVIDIA 40G A100 GPU. The training times for different tasks ranged from just over ten minutes (RTE) to more than ten hours (QQP).

### A.4.2 EXPERIMENTS WITH SELF-IMPLEMENTATION

We further find that the increasing wall time and the GPU memory usage come from the public library tutel we used to implement EMoE. We reimplement our method and observed that EMoE does not require significant additional run time and memory usage. Specifically, we introduce an alternative implementation approach in EMoE where hidden states are used to calculate gate scores after computing the first layer. These scores mask the outputs of unselected experts, mimicking the effect of MoEs. Though this theoretically increases FLOPS compared to traditional MoEs, in practice, the speed is comparable to dense models, as demonstrated in Tables 7, 8 and 9.

### A.5 TABULAR RESULTS

### A.5.1 FULL TABLES IN FULL FINE-TUNING WITH STANDARD DEVIATION

In this section, we present the mean and variance of experiments conducted with three different random seeds. The Domainbed results are demonstrated in Table 13, full fine-tuning results are in Table 14, LoRA tuning results are in Table 15.

Table 13: Overall out-of-domain performances with different selection criteria. All the reported results are obtained from three independent experiments conducted following the Domainbed code-base. The best result is highlighted in **bold**. In cases where results are the same, the best result is determined by the smallest standard deviation. EMoE demonstrates comparable results to GMoE.

Results with ViT-small backbone

| Algorithm | PACS | VLCS | OfficeHome | TerraInc | Avg |
|---|---|---|---|---|---|
| train-validation selection criterion | | | | | |
| ViT | 86.9±0.2 | **79.7±0.4** | 73.0±0.2 | 44.0±1.1 | 70.90 |
| GMoE | 87.7±0.2 | 79.6±0.4 | 73.1±0.3 | 45.4±0.3 | 71.45 |
| EMoE-learn | 87.2±0.4 | 79.6±0.2 | 72.5±0.2 | **46.1±0.4** | 71.35 |
| EMoE | **87.8±0.2** | 79.5±0.4 | **73.1±0.2** | 45.9±0.3 | **71.58** |
| leave-one-domain-out selection criterion | | | | | |
| ViT | 86.1±0.6 | 79.7±0.4 | **73.3±0.1** | 45.0±0.5 | 71.03 |
| GMoE | 86.5±0.3 | 80.5±0.2 | 73.1±0.3 | 45.3±0.6 | 71.35 |
| EMoE-learn | **86.8±0.0** | 79.6±0.3 | 72.6±0.2 | 45.8±0.6 | 71.20 |
| EMoE | 86.8±0.1 | **80.6±0.4** | 73.3±0.2 | **46.1±0.6** | **71.70** |
| test-domain selection criterion | | | | | |
| ViT | 86.5±0.4 | 78.2±0.4 | **73.1±0.2** | 44.0±0.5 | 70.45 |
| GMoE | 87.2±0.4 | 79.0±0.3 | **73.4±0.2** | 45.3±0.4 | 71.23 |
| EMoE-learn | 87.4±0.2 | **79.1±0.3** | 72.8±0.1 | 45.4±0.6 | 71.18 |
| EMoE | **87.6±0.5** | 79.0±0.2 | 73.3±0.0 | **45.5±0.1** | **71.35** |

Results with ViT-base backbone

| Algorithm | PACS | VLCS | OfficeHome | TerraInc | Avg |
|---|---|---|---|---|---|
| train-validation selection criterion | | | | | |
| ViT | 89.1±0.0 | **80.7±0.1** | 77.2±0.1 | 47.3±0.8 | 73.58 |
| GMoE | **90.0±0.3** | 80.4±0.6 | 77.0±0.1 | **49.2±1.1** | **74.15** |
| EMoE-learn | 89.8±0.2 | 80.6±0.2 | 76.5±0.1 | 48.7±0.5 | 73.90 |
| EMoE | 89.4±0.4 | 80.7±0.2 | **77.3±0.1** | 48.5±0.5 | 73.98 |
| leave-one-domain-out selection criterion | | | | | |
| ViT | 88.9±0.4 | 80.8±0.3 | **77.5±0.1** | 46.1±0.6 | 73.33 |
| GMoE | 89.3±0.6 | 81.0±0.3 | 76.7±0.1 | 50.1±1.1 | 74.28 |
| EMoE-learn | 89.3±0.2 | 81.2±0.1 | 76.5±0.1 | **50.5±0.2** | 74.38 |
| EMoE | **89.6±0.2** | **81.6±0.2** | 77.4±0.1 | 50.0±1.1 | **74.65** |
| test-domain selection criterion | | | | | |
| ViT | 88.8±0.7 | 79.0±0.5 | 77.2±0.0 | 46.7±0.4 | 72.93 |
| GMoE | 89.7±0.5 | 79.0±0.3 | 77.0±0.1 | **48.8±0.4** | 73.63 |
| EMoE-learn | **89.7±0.4** | **79.7±0.2** | 76.6±0.1 | 48.7±0.3 | 73.68 |
| EMoE | **89.7±0.4** | **79.7±0.2** | **77.5±0.1** | 48.8±0.6 | **73.93** |

### A.5.2 FULL TABLES AND FIGURE DATA SOURCES IN ANALYSIS

In the Analysis section 5.1, for the sake of clarity, we have transformed tabular data into graphs or retained only a subset of the results. The original and complete results corresponding to them are presented in this section. Figure 3 is summarized from Table 16 and Table 17. The OOD results in Table 3 are from 15. The ablation studies are from Table 19 and Table 20.

### A.6 MORE VISUALIZATION RESULTS

In this part, we demonstrate more gating visualization results on SST-2, STSB, MRPC, and RTE in Figure 8. These results are consistent with earlier findings: (1) Both avg-k gating and learned gating

Table 14: Results on GLUE tasks with different backbones and algorithms. All the reported results are obtained from 3 independent experiments. The average accuracy (avg) is reported along with the relative improvement compared to the baseline. The best result is highlighted in **bold**.

| Backbone | Algorithm | MRPC | CoLA | RTE | STSB | SST2 | Avg |
|---|---|---|---|---|---|---|---|
| BERT-B | baseline | 88.45±0.40 | 60.67±0.54 | 68.95±0.69 | 87.87±0.12 | 91.97±0.19 | 79.582 |
| | noisy tuning | 88.43±0.12 | 61.79±0.16 | 71.36±0.17 | 88.27±0.94 | 92.32±0.25 | 80.43(+0.85) |
| | GMoE | 88.63±0.53 | 61.25±2.36 | 70.28±0.68 | 88.63±0.65 | 92.28±0.24 | 80.21(+0.63) |
| | EMoE-learn | 89.05±0.23 | **62.46±1.01** | 70.40±1.28 | 88.47±0.74 | 92.58±0.14 | **80.59(+1.01)** |
| | EMoE | **89.45±0.36** | 61.55±0.67 | 69.68±1.02 | **88.71±0.50** | **92.89±0.19** | 80.46(+0.87) |
| BERT-L | baseline | 89.82±1.30 | 65.41±0.47 | 74.89±1.39 | 89.87±0.28 | 93.50±0.24 | 82.70 |
| | noisy tuning | 90.42±0.35 | 64.75±1.31 | 73.41±1.62 | 90.05±0.46 | 93.65±0.11 | 82.46(-0.24) |
| | GMoE | **91.24±0.25** | 64.90±1.26 | 74.24±1.04 | 90.00±0.64 | 93.58±0.25 | 82.79(+0.09) |
| | EMoE-learn | 90.57±0.43 | 65.51±0.32 | 74.72±2.13 | 90.22±0.49 | **93.73±0.35** | 82.95(+0.25) |
| | EMoE | 90.74±0.65 | **65.79±1.16** | **76.17±0.00** | **90.31±0.43** | 93.58±0.32 | **83.32(+0.62)** |
| GPT2 | baseline | 84.46±0.51 | 47.07±1.60 | 67.15±0.51 | 86.29±0.29 | 92.13±0.30 | 75.42 |
| | noisy tuning | 84.15±0.92 | 46.16±2.79 | 67.51±0.78 | 86.09±0.38 | 92.13±0.27 | 75.21(-0.21) |
| | GMoE | 85.07±0.45 | 47.77±3.20 | 67.51±0.51 | 86.57±0.29 | 92.35±0.35 | 75.85(+0.43) |
| | EMoE-learn | **85.73±0.09** | 47.24±1.48 | 67.99±0.17 | **86.66±0.32** | 92.35±0.38 | 75.99(+0.57) |
| | EMoE | 85.40±0.77 | **48.00±1.50** | 68.95±0.29 | 86.64±0.16 | **92.70±0.22** | **76.34(+0.92)** |

Table 15: Results on various algorithms with different models and tasks. All the reported results are obtained from 3 independent experiments. OOD Metrics (averaged over 14 OOD tasks, lower is better) provide additional information for out-of-distribution generalization. The best result is highlighted in **bold**, and the second is marked with underline.

| Algorithm | MRPC | CoLA | RTE | STSB | SST2 | QNLI | QQP | MNLI | ID-Avg | OOD |
|---|---|---|---|---|---|---|---|---|---|---|
| BERT-Large (340 Million Parameters) Results | | | | | | | | | | |
| LoRA | 89.97±0.40 | 63.40±0.62 | 72.92±1.64 | 90.51±0.18 | 93.16±0.19 | 92.20±0.13 | 87.21±0.60 | 85.40±0.07 | 84.35 | 4.86 |
| Block | 89.34±0.84 | 62.10±0.91 | 71.96±1.68 | 90.39±0.14 | 93.35±0.43 | 92.04±0.16 | 88.45±0.07 | 86.20±0.10 | 84.23(-0.12) | 4.95 |
| Block+GMoE | 89.45±0.72 | 63.80±0.71 | 72.56±0.29 | 90.29±0.07 | 93.85±0.11 | 92.32±0.14 | 87.99±0.06 | 85.92±0.13 | 84.52(+0.18) | 4.04 |
| Block+EMoE-learn | 89.79±0.23 | 64.16±0.87 | 73.16±1.04 | 90.27±0.03 | 93.85±0.11 | 92.40±0.06 | 88.01±0.12 | 85.76±0.19 | 84.68(+0.33) | 3.94 |
| Block+EMoE | 89.77±0.46 | 63.25±0.50 | 71.60±0.68 | 90.31±0.09 | 93.69±0.32 | 92.09±0.13 | 88.08±0.19 | 86.21±0.16 | 84.38(+0.03) | 5.89 |
| EMoE | 90.85±0.61 | 65.33±0.40 | 75.21±1.62 | 90.43±0.06 | 93.50±0.33 | 92.23±0.10 | 87.74±0.10 | 85.43±0.10 | 85.09(+0.74) | 4.37 |
| EMoE+LN | 90.47±0.33 | 64.39±0.31 | 73.41±1.04 | 90.54±0.03 | 93.00±0.16 | 92.31±0.05 | 88.79±0.17 | 85.50±0.10 | 84.80(+0.46) | 4.53 |
| EMoE-learn | 89.87±0.50 | 64.00±0.57 | 71.36±1.39 | 90.48±0.10 | 93.65±0.33 | 92.40±0.11 | 87.55±0.14 | 85.62±0.23 | 84.37(+0.02) | 4.66 |
| EMoE-learn+LN | 89.9±0.25 | 64.16±1.16 | 72.44±0.45 | 90.45±0.10 | 93.42±0.38 | 92.15±0.10 | 87.70±0.04 | 85.52±0.24 | 84.47(0.12) | 4.28 |
| GPT2-XL (1.5 Billion Parameters) Results | | | | | | | | | | |
| LoRA | 86.83±0.87 | 60.88±2.54 | 78.70±0.59 | 89.07±0.11 | 95.18±0.28 | 91.84±0.09 | 87.41±1.74 | 86.93±0.15 | 84.61 | 5.61 |
| Block | 86.59±1.45 | 61.18±1.74 | 79.78±2.22 | 89.08±0.15 | 95.45±0.19 | 91.88±0.05 | 87.71±2.95 | 86.95±0.08 | 84.83(+0.22) | 5.13 |
| Block+GMoE | 87.02±0.76 | 62.81±1.51 | 79.78±1.35 | 89.21±0.20 | 95.41±0.28 | 92.18±0.11 | 89.10±0.78 | 87.17±0.20 | 85.34(+0.73) | 4.33 |
| Block+EMoE-learn | 87.31±1.23 | 62.24±1.51 | 79.54±0.17 | 89.33±0.11 | 95.30±0.09 | 92.20±0.09 | 88.59±1.68 | 87.06±0.18 | 85.20(+0.59) | 4.05 |
| Block+EMoE | 87.86±0.98 | 62.88±0.54 | 80.05±0.29 | 89.18±0.25 | 95.49±0.39 | 92.10±0.15 | 89.69±0.15 | 86.87±0.11 | 85.52(+0.91) | 5.71 |
| EMoE | 87.75±0.14 | 62.27±0.93 | 80.02±0.34 | 89.37±0.30 | 95.41±0.32 | 92.10±0.15 | 89.58±0.10 | 87.06±0.25 | 85.45(+0.84) | 3.88 |
| EMoE+LN | 88.05±0.35 | 63.11±0.51 | 79.90±1.51 | 89.40±0.22 | 95.18±0.28 | 92.23±0.11 | 89.70±0.09 | 87.03±0.14 | 85.58(+0.97) | 4.39 |
| EMoE-learn | 87.93±0.61 | 61.50±1.09 | 79.90±0.61 | 89.48±0.24 | 95.18±0.11 | 92.33±0.093 | 89.71±0.06 | 87.00±0.19 | 85.38(+0.77) | 4.40 |
| EMoE-learn+LN | 87.04±1.11 | 62.64±0.84 | 79.78±0.59 | 89.50±0.22 | 95.30±0.50 | 92.31±0.19 | 89.43±0.35 | 87.00±0.12 | 85.38(+0.77) | 3.67 |

converge, as indicated by the lower halves of the plots. (2) avg-k gating is more stable than learned gating. This could mitigate data inefficiency resulting from inconsistencies in gating across different stages of training (Zuo et al., 2022).

Table 16: ID and OOD results of BERT-L for different settings. "LoRA-to-EMoE" refers to converting a model tuned using standard LoRA into EMoE for testing. On the other hand, "EMoE-to-LoRA" involves merging a tuned EMoE model back into a standard dense model during testing.

| Algorithm | CoLA | Gram | MRPC | Twitter | RTE | Hans | SciTail | STSB | Sick | Avg |
|---|---|---|---|---|---|---|---|---|---|---|
| LoRA | 60.89±2.55 | 41.77±1.62 | 86.83±0.87 | 75.42±2.71 | 78.70±1.02 | 60.37±1.31 | 77.36±0.73 | 89.22±0.13 | 78.48±0.33 | 72.12 |
| LoRA-to-EMoE | 61.31±2.14 | 41.99±1.56 | 86.83±0.91 | 75.15±3.02 | 78.58±0.95 | 60.39±1.32 | 77.24±0.68 | 89.23±0.13 | 78.53±0.35 | 72.14 |
| EMoE | 62.69±0.91 | 42.95±0.95 | 87.82±0.17 | 76.07±2.12 | 79.54±0.45 | 61.56±1.65 | 78.09±0.56 | 89.39±0.31 | 78.57±0.67 | 72.96 |
| EMoE-to-LoRA | 62.69±0.91 | 42.94±0.95 | 87.82±0.17 | 76.06±2.12 | 79.54±0.45 | 61.56±1.65 | 78.07±0.58 | 89.39±0.3 | 78.58±0.67 | 72.96 |

Table 17: ID results of BERT-L for different settings. "Cluster-top" refers to EMoE utilizing avg-k gating. "Cluster-not-top" represents a scenario where, during gating, the top-k experts are removed. Similarly, "Cluster-bottom" involves selecting the bottom-k experts with the lowest scores during gating. "Random" denotes the approach of randomly selecting key values to construct experts. The terms "top," "not-top," and "bottom" have the same meanings as in the cluster situations.

| Algorithm | MRPC | CoLA | RTE | STSB | SST2 | QNLI | Avg |
|---|---|---|---|---|---|---|---|
| LoRA | 89.97±0.40 | 63.40±0.62 | 72.92±1.64 | 90.51±0.18 | 93.16±0.19 | 92.20±0.13 | 83.69 |
| Cluster-top | 90.85±0.61 | 65.33±0.40 | 75.21±1.62 | 90.54±0.03 | 93.50±0.33 | 92.23±0.10 | 84.61(+0.92) |
| Cluster-not-top | 89.61±0.76 | 63.21±0.44 | 72.56±1.28 | 90.31±0.07 | 93.12±0.34 | 92.14±0.18 | 83.49(-0.20) |
| Cluster-bottom | 89.21±0.69 | 63.08±1.09 | 71.72±0.34 | 90.15±0.18 | 92.97±0.19 | 92.13±0.31 | 83.21(-0.48) |
| Random-top | 89.88±0.75 | 63.26±0.39 | 72.56±1.06 | 90.33±0.05 | 93.35±0.00 | 92.14±0.20 | 83.59(-0.11) |
| Random-not-top | 90.09±0.75 | 63.35±0.34 | 72.44±1.19 | 90.44±0.07 | 93.31±0.25 | 92.20±0.16 | 83.64(-0.05) |
| Random-bottom | 89.47±0.23 | 63.17±0.98 | 71.96±0.74 | 90.30±0.23 | 93.11±0.25 | 92.10±0.11 | 83.35(-0.34) |

Table 18: Raw OOD performances across 13 tasks. Average results with standard deviation and best results are reported separately. Due to the large deviation across seeds overall methods, we use Friedman rank metrics (Friedman, 1940).

| Algorithm | Twitter-M | GrammarTest | Hans | SciTail | Sick-S | NewsQA | Amazon | Flipkart | Imdb | Yelp | MRPC | Twitter-Q | Sick-M |
|---|---|---|---|---|---|---|---|---|---|---|---|---|---|
| **BERT-Large (340 Million Parameters) Results (mean and standard deviation)** | | | | | | | | | | | | | |
| LoRA | 80.09±0.49 | 43.55±1.34 | 55.51±1.94 | 81.13±0.88 | 81.63±0.16 | 78.22±0.17 | 89.06±1.81 | 91.79±0.27 | 84.97±1.32 | 88.40±2.06 | 71.73±0.46 | 78.10±0.33 | 53.17±0.92 |
| LoRA+block | 80.70±0.35 | 42.94±0.47 | 54.35±1.08 | 80.99±0.33 | 81.58±0.18 | 78.33±0.24 | 88.21±1.20 | 91.26±0.96 | 84.10±0.85 | 89.13±0.20 | 71.57±0.87 | 79.22±0.59 | 53.67±0.77 |
| GMoE | 81.17±0.54 | 46.76±1.91 | 59.72 | 79.41±0.10 | 81.35±0.34 | 78.05±0.28 | 89.27±1.38 | 91.77±0.07 | 84.93±1.09 | 88.70±1.77 | 71.08±0.69 | 78.70±0.37 | 53.85±1.27 |
| Block+EMoE-learn | 81.09±0.47 | 43.29±1.42 | 57.55±3.48 | 77.72±0.61 | 80.78±0.49 | 78.32±0.13 | 90.17±0.32 | 91.82±0.18 | 85.71±0.19 | 89.62±0.24 | 72.71±0.23 | 77.85±1.03 | 54.69±0.15 |
| Block+EMoE | 81.07±0.53 | 46.33±1.40 | 61.30±2.24 | 79.62±0.45 | 80.15±0.61 | 79.03±0.25 | 88.81±0.19 | 91.33±0.23 | 84.92±0.07 | 88.87±0.26 | 70.59±1.00 | 74.62±2.70 | 53.04±1.82 |
| EMoE | 80.82±0.10 | 44.18±0.63 | 64.31 | 76.93±2.52 | 81.19±0.27 | 77.87±0.19 | 90.25±0.10 | 92.14±0.25 | 85.63±0.30 | 90.21±0.14 | 71.24±1.22 | 78.00±0.87 | 51.55±0.88 |
| Block+EMoE+ln | 81.04±0.10 | 45.05±0.88 | 60.73 | 77.92±1.64 | 81.27±0.17 | 77.40±0.25 | 90.08±0.07 | 91.64±0.54 | 85.67±0.29 | 89.23±0.14 | 72.14±0.76 | 77.34±1.33 | 52.81±0.48 |
| EMoE+ln | 81.08±0.24 | 45.71 | 58.11 | 79.84 | 81.25±0.13 | 77.63 | 90.15 | 92.33 | 86.01 | 90.60 | 72.79 | 79.53 | 53.28 |
| EMoE-learn | 81.34 | 48.75 | 59.09 | 77.52±2.15 | 81.25 | 78.16±0.30 | 89.84±0.13 | 91.87±0.28 | 84.97±0.33 | 89.76±0.33 | 71.32±2.03 | 77.47±0.76 | 53.29±0.45 |
| EMoE-learn+ln | 81.34±0.30 | 44.46±1.67 | 59.09 | 77.78±1.71 | 81.13±0.26 | 78.28±0.13 | 90.02±0.29 | 91.48±0.30 | 85.62±0.46 | 89.87±0.30 | 71.90±0.61 | 77.38±1.77 | 53.44±0.47 |
| **BERT-Large (340 Million Parameters) Results (best result)** | | | | | | | | | | | | | |
| LoRA | 80.75 | 45.42 | 58.24 | 82.03 | 81.79 | 78.46 | 90.35 | 92.05 | 85.95 | 90.37 | 72.06 | 78.41 | 54.44 |
| LoRA+block | 81.19 | 43.56 | 55.48 | 81.45 | 81.83 | 78.63 | 89.49 | 92.48 | 85.25 | 89.17 | 72.79 | 79.88 | 54.71 |
| GMoE | 81.78 | 49.14 | 59.72 | 79.55 | 81.66 | 78.41 | 90.31 | 91.86 | 85.93 | 90.42 | 72.06 | 79.02 | 55.16 |
| Block+EMoE-learn | 81.72 | 44.45 | 61.70 | 78.47 | 81.38 | 78.46 | 90.59 | 92.04 | 85.93 | 89.95 | 73.04 | 78.89 | 54.91 |
| Block+EMoE | 81.49 | 47.77 | 54.57 | 80.15 | 80.85 | 79.32 | 88.98 | 91.66 | 85.00 | 89.23 | 71.81 | 77.14 | 55.56 |
| EMoE | 80.95 | 45.05 | 64.31 | 79.64 | 81.55 | 78.10 | 90.15 | 92.42 | 86.03 | 90.34 | 72.55 | 79.23 | 52.24 |
| Block+EMoE+ln | 81.17 | 45.71 | 60.73 | 79.84 | 81.48 | 77.63 | 90.15 | 92.33 | 85.67 | 90.21 | 72.79 | 79.53 | 53.28 |
| EMoE+ln | 81.34 | 48.75 | 58.11 | 79.89 | 81.25 | 78.58 | 89.93 | 92.23 | 85.38 | 90.21 | 72.79 | 79.53 | 53.91 |
| EMoE-learn | 81.72 | 45.71 | 60.73 | 79.84 | 81.48 | 78.76 | 89.84 | 92.47 | 84.49 | 90.77 | 74.02 | 78.33 | 53.91 |
| EMoE-learn+ln | 81.72 | 46.82 | 59.09 | 79.60 | 81.41 | 78.47 | 90.27 | 91.72 | 86.11 | 90.17 | 72.55 | 78.67 | 54.10 |
| **GPT2-XL (1.5 Billion Parameters) Results (mean and standard deviation)** | | | | | | | | | | | | | |
| LoRA | 75.42±2.71 | 41.78±1.62 | 60.37±1.32 | 77.36±0.73 | 78.48±0.33 | 78.57±0.59 | 89.91±0.73 | 90.05±0.44 | 85.57±1.34 | 88.85±0.38 | 68.22±1.27 | 73.60±2.87 | 57.39±0.34 |
| LoRA+block | 76.87±2.47 | 41.09±1.80 | 58.67±1.23 | 78.57±1.23 | 77.66±0.25 | 78.78±0.55 | 90.11±0.71 | 91.54±0.94 | 83.54±0.57 | 88.85±1.01 | 69.53±0.64 | 68.22±7.84 | 58.53±0.46 |
| GMoE | 75.28±3.67 | 42.50±1.37 | 62.59±0.37 | 77.89±0.56 | 78.33±0.58 | 79.18±0.04 | 90.11±0.38 | 88.64±4.52 | 83.23±0.40 | 89.47±0.26 | 70.10±1.39 | 74.61±3.71 | 58.29±0.78 |
| Block+EMoE-learn | 74.80±5.15 | 44.15±1.95 | 61.67±1.18 | 78.18±1.04 | 78.27±0.21 | 78.97±0.20 | 90.22±0.59 | 91.04±0.65 | 83.82±0.54 | 89.49±0.67 | 69.93±0.50 | 73.41±3.94 | 57.58±0.57 |
| Block+EMoE | 75.37±3.04 | 39.33±2.21 | 58.32±2.40 | 72.05±3.04 | 77.70±0.59 | 79.32±0.06 | 90.24±0.36 | 91.79±0.27 | 83.79±0.27 | 89.23±0.38 | 63.24±8.56 | 70.60±2.89 | 56.85±0.57 |
| EMoE | 76.07±2.12 | 42.95±0.95 | 61.56±1.65 | 78.09±0.56 | 78.57±0.67 | 78.87±0.38 | 90.39±0.55 | 91.87±0.22 | 83.55±1.04 | 89.14±1.39 | 67.73±1.55 | 74.06±4.79 | 57.33±0.59 |
| Block+EMoE+ln | 74.35±3.50 | 42.24±0.89 | 61.88±2.45 | 78.32±0.95 | 78.67±0.62 | 79.02±0.30 | 89.51±0.90 | 91.19±0.56 | 83.12±1.90 | 89.29±1.09 | 69.20±1.03 | 72.74±5.10 | 57.62±0.02 |
| EMoE+ln | 74.61±4.75 | 40.50±0.46 | 59.50±3.74 | 80.04±0.46 | 78.27±0.59 | 79.06±0.50 | 90.59±0.66 | 91.69±0.59 | 83.23±1.54 | 89.79±1.35 | 67.48±1.31 | 73.62±1.12 | 56.22±0.15 |
| EMoE-learn | 74.50±4.16 | 42.52±0.41 | 58.90±4.16 | 79.56±0.81 | 78.27±0.54 | 79.19±0.32 | 90.23±0.86 | 91.98±0.35 | 83.31±1.08 | 88.50±1.92 | 73.62±1.12 | 74.77 | 56.42 |
| EMoE-learn+ln | 77.17±1.58 | 42.52±0.41 | 58.90±4.16 | 79.56±0.81 | 78.27±0.54 | 79.19±0.32 | 90.23±0.86 | 91.98±0.35 | 83.31±1.08 | 88.50±1.92 | 68.63±1.51 | 71.26±3.49 | 56.69±0.37 |
| **GPT2-XL (1.5 Billion Parameters) Results (best result)** | | | | | | | | | | | | | |
| LoRA | 78.17 | 43.93 | 61.77 | 78.39 | 78.87 | 79.12 | 90.52 | 91.03 | 84.81 | 89.39 | 69.12 | 77.48 | 57.79 |
| LoRA+block | 80.17 | 43.09 | 59.52 | 79.45 | 78.33 | 79.42 | 90.94 | 92.87 | 84.12 | 90.28 | 70.10 | 74.55 | 57.64 |
| GMoE | 80.23 | 43.96 | 63.05 | 78.68 | 78.00 | 79.33 | 90.57 | 92.27 | 83.69 | 89.81 | 71.08 | 79.79 | 59.36 |
| Block+EMoE-learn | 80.04 | 46.01 | 62.74 | 79.65 | 79.06 | 79.24 | 90.67 | 91.94 | 84.45 | 90.42 | 70.59 | 78.78 | 58.28 |
| Block+EMoE | 79.02 | 41.55 | 61.64 | 76.30 | 78.33 | 79.40 | 90.68 | 92.00 | 84.12 | 89.60 | 73.31 | 73.31 | 57.64 |
| EMoE | 77.58 | 43.84 | 63.78 | 78.75 | 79.29 | 79.26 | 91.17 | 92.19 | 84.42 | 90.22 | 69.85 | 80.81 | 58.01 |
| Block+EMoE+ln | 77.92 | 43.39 | 65.32 | 79.16 | 79.25 | 79.35 | 90.48 | 91.74 | 85.73 | 90.79 | 70.59 | 79.81 | 57.87 |
| EMoE+ln | 79.35 | 41.15 | 64.56 | 80.40 | 78.76 | 79.76 | 91.32 | 92.47 | 84.49 | 90.77 | 69.12 | 74.77 | 56.42 |
| EMoE-learn | 79.35 | 43.01 | 64.32 | 80.57 | 79.02 | 79.63 | 91.40 | 92.41 | 84.84 | 90.57 | 70.34 | 74.77 | 57.13 |

Table 19: Results of different MoEs Configurations

| Algorithm | MRPC | CoLA | RTE | STSB | Avg |
|---|---|---|---|---|---|
| Dense | $86.83_{\pm 0.87}$ | $60.88_{\pm 2.54}$ | $78.70_{\pm 0.59}$ | $89.07_{\pm 0.11}$ | 78.87 |
| EMoE | $88.05_{\pm 0.35}$ | $63.11_{\pm 0.21}$ | $80.02_{\pm 0.34}$ | $89.37_{\pm 0.24}$ | 80.14 |
| EMoE-learn | $87.93_{\pm 0.61}$ | $62.87_{\pm 0.71}$ | $79.90_{\pm 0.61}$ | $89.40_{\pm 0.08}$ | 80.03 |
| EMoE-last-every2 | $87.27_{\pm 0.47}$ | $61.60_{\pm 0.63}$ | $79.18_{\pm 0.17}$ | $89.38_{\pm 0.24}$ | 79.36 |
| EMoE-learn-learn-every2 | $87.26_{\pm 0.21}$ | $61.82_{\pm 1.10}$ | $78.46_{\pm 1.22}$ | $89.31_{\pm 0.15}$ | 79.21 |
| EMoE-every2 | $86.78_{\pm 0.34}$ | $59.21_{\pm 0.79}$ | $77.38_{\pm 1.04}$ | $89.31_{\pm 0.06}$ | 78.17 |
| EMoE-learn-every2 | $86.71_{\pm 1.32}$ | $54.02_{\pm 0.47}$ | $74.25_{\pm 0.74}$ | $88.51_{\pm 0.31}$ | 75.87 |

Table 20: Comparison of EMoE and Dense Results with Different Training Data Fraction

| Data Fraction | CoLA | | MRPC | | RTE | | STSB | | SST2 | | QNLI | | Average Diff |
|---|---|---|---|---|---|---|---|---|---|---|---|---|---|
| | EMoE | Dense | EMoE | Dense | EMoE | Dense | EMoE | Dense | EMoE | Dense | EMoE | Dense | |
| 1.0 | 62.27 | 60.88 | 87.75 | 86.83 | 80.02 | 78.70 | 89.37 | 89.07 | 95.41 | 95.18 | 92.10 | 91.84 | 0.74 |
| 0.9 | 61.58 | 60.01 | 87.52 | 86.49 | 79.87 | 77.85 | 89.18 | 89.07 | 95.41 | 95.16 | 92.05 | 91.94 | 0.85 |
| 0.8 | 60.89 | 59.28 | 86.58 | 86.35 | 79.22 | 76.77 | 88.98 | 88.99 | 95.34 | 95.06 | 91.94 | 91.83 | 0.78 |
| 0.7 | 59.29 | 58.25 | 86.10 | 85.56 | 77.98 | 76.77 | 87.95 | 87.95 | 95.41 | 95.06 | 92.04 | 91.74 | 0.57 |
| 0.6 | 58.91 | 57.93 | 85.61 | 84.76 | 76.29 | 75.45 | 86.70 | 86.17 | 95.26 | 95.03 | 91.83 | 91.54 | 0.62 |
| 0.5 | 55.18 | 53.89 | 84.91 | 84.76 | 76.53 | 75.21 | 85.63 | 85.59 | 95.19 | 94.91 | 91.23 | 91.12 | 0.53 |
| 0.3 | 50.17 | 50.29 | 82.80 | 82.59 | 73.52 | 72.44 | 80.23 | 79.08 | 94.82 | 94.72 | 90.45 | 90.26 | 0.44 |
| 0.1 | 46.47 | 45.54 | 78.17 | 77.85 | 63.05 | 63.17 | 62.33 | 60.81 | 94.49 | 94.38 | 88.39 | 88.32 | 0.47 |

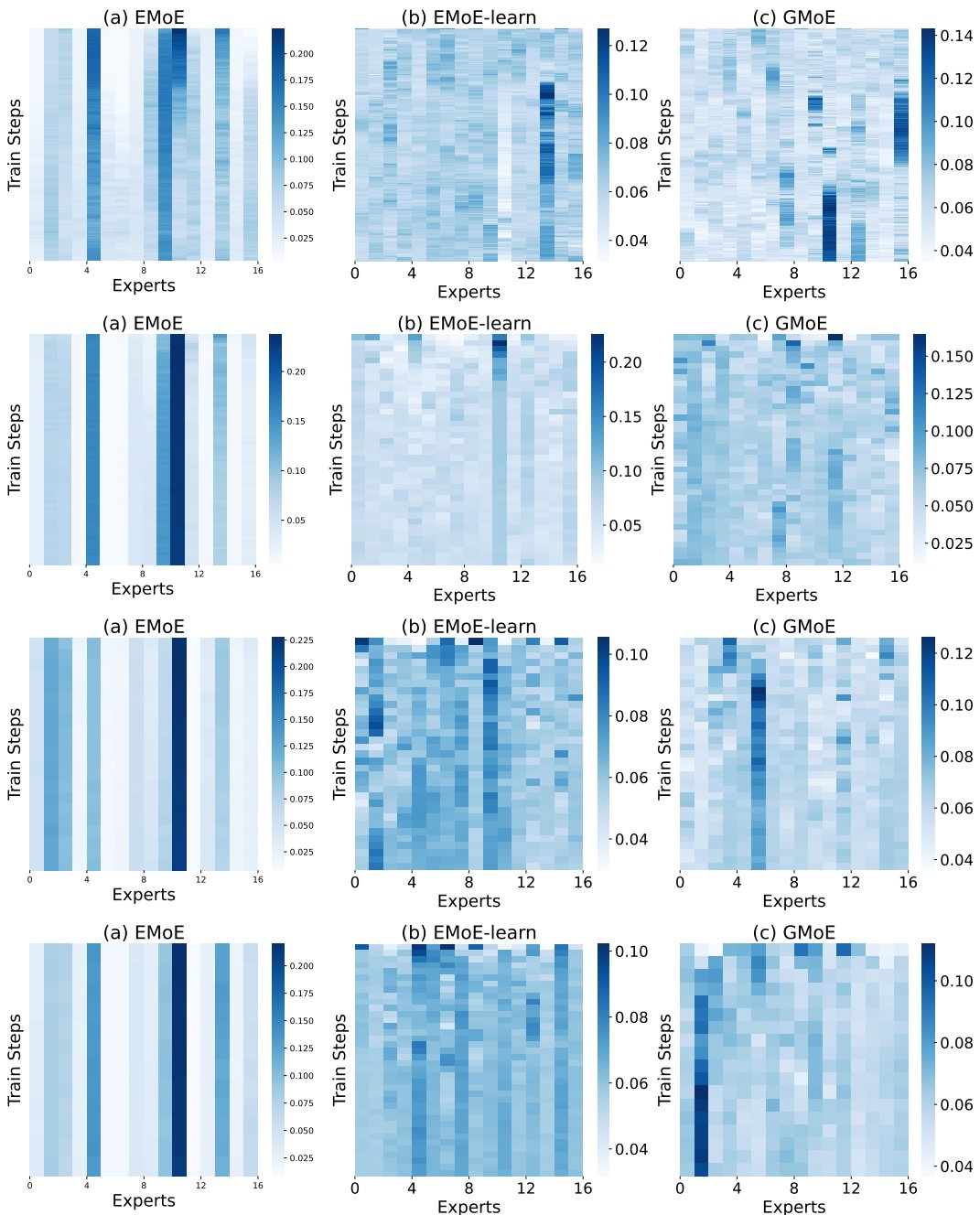

Figure 8: Expert selections during training with distinct gating functions (avg-k vs learned gate) and expert types (modules from dense vs repetitions of dense). The vertical axis illustrates training steps, with top-down arrangement signifying begin-end; the horizontal axis represents expert selection frequency within 1K steps. (a), (b) and (c) correspond respectively to EMoE, EMoE-learn, and GMoE configurations. The subplots from top to bottom are results for SST-2, STS-B, MRPC, and RTE.

