# OpenReview forum: "Emergent Mixture-of-Experts: Can Dense Pre-trained Transformers Benefit from Emergent Modular Structures?"
_ICLR.cc/2024/Conference — ICLR 2024 Conference Withdrawn Submission_

### Official Review · Reviewer_wdQa · 2023-10-23

**Soundness:** 4 excellent
**Presentation:** 3 good
**Contribution:** 3 good
**Rating:** 6
**Confidence:** 4

**Summary:**

This paper explores whether dense pre-trained transformers can benefit from emergent modular structures, which are spontaneously formed during the early pre-training phase. The authors propose Emergent Mixture-of-Experts (EMoE), a method that externalizes the emergent modularity into explicit Mixture-of-Experts without introducing additional parameters. EMoE is evaluated on various downstream tasks, models, and configurations, with a total of 1785 models fine-tuned. The results show that EMoE effectively boosts in-domain and out-of-domain generalization abilities, mitigates negative knowledge transfer, and is robust to various configurations. It also demonstrates higher data efficiency of up to 20% compared to the vanilla fine-tuning and competitive performance with the strong baseline GMoE.

**Strengths:**

1. This paper appears to be the first to systematically investigate the emergent modular structures within the Feed-Forward Networks (FFNs) of pre-trained Transformers, contributing valuable insights to the field.

2. The authors propose a straightforward yet effective method for Mixture-of-Experts (MoE) initialization that outperforms existing solutions across both vision and natural language processing tasks, demonstrating its practicality and usefulness.

3. The experimental setup is comprehensive, with a total of 1785 models fine-tuned, showcasing the thoroughness of the investigation and the robustness of the proposed method in various scenarios.

**Weaknesses:**

1. The choice of N and k in EMoE differs significantly from existing Mixture-of-Experts (MoEs) models, which may potentially diminish some of the advantages typically associated with MoEs. It would be beneficial to further explore the implications of these choices and their impact on the performance of the method. (Refer to Question 1)

2. Some aspects of the experimental settings could be clarified or expanded upon to ensure a thorough understanding of the results and their significance. (Refer to Question 2, 3)

3. The presentation of results in tables and figures could be improved to enhance readability and better convey the key findings of the paper. Clearer visual representations would help readers grasp the main takeaways more effectively. (Refer to Question 4, 5)

**Questions:**

1. In previous MoE works, k is often set to {1, 2} to achieve high time efficiency, while the value of k in EMoE is {16, 32}. How does this choice affect time efficiency? It would be helpful if the authors could provide a comparison in terms of GMACs for Transformers, MoEs, GMoEs, and EMoEs.

2. The results demonstrate that EMoE outperforms EMoE-learn when the number of fine-tune iterations is the same. This might be attributed to the gate of EMoE having a better initialization point and converging faster. It would be interesting to see the performance of EMoE-learn with longer finetune iterations compared to EMoE.

3. In GMoEs, auxiliary loss functions (as described in Appendix C.4 of the GMoE paper) are employed to achieve good performance. Were these loss functions used for the GMoE baselines in this study? Additionally, could these loss functions improve the performance of EMoE?

4. The font size of "Office" and "Terra" in the figures appears to be smaller than that of the other labels. It would be helpful to ensure consistency in font sizes across all labels.

5. The right figure in Figure 4 seems to lack data points. Could the authors clarify if this is intentional or if there might be an issue with the figure's presentation?

---

> ### Author Response · Authors · 2023-11-17
> **Response to wdQa**
>
> > *1. In previous MoE works, $k$ is often set to {1, 2} to achieve high time efficiency, while the value of $k$ in EMoE is {16, 32}. How does this choice affect time efficiency? It would be helpful if the authors could provide a comparison in terms of GMACs for Transformers, MoEs, GMoEs, and EMoEs.*
>
> Thanks for your question. Increasing the number of experts $N$ and the selected expert $k$ for EMoE does not severely affect the time efficiency and computation cost. The key difference between EMoE and previous MoEs works (including GMoE) is that EMoE split the original FFNs layer into different experts. The larger the number of experts, the smaller the expert size. As a result, the computational efficiency is largely independent of the EMoE's settings (different $k$ and $N$).  Table 3, 4, and 5 in General Response shows no difference in FLOPS and time between the scenarios with k set to 16 (9.12e16) and 32 (9.24e16). Neither setting significantly differs from the standard Llama2-7B (8.97e16). On Llama2-7B, the introduction of EMoE (8.99e17) has minimal difference relative to standard LoRA Tuning (8.93e17).
>
> When it comes to GMoE, larger $k$ will lead to computation overhead because they replicate the original FFNs from MoEs.  Large $k$ gets more experts involved, and every expert has the same size as the original FFNs layer, leading to computation overhead. We will add relevant discussions in the final version of our paper.
>
> > 2. *The results demonstrate that EMoE outperforms EMoE-learn when the number of fine-tuning iterations is the same. This might be attributed to the gate of EMoE having a better initialization point and converging faster. It would be interesting to see the performance of EMoE-learn with longer finetune iterations compared to EMoE.*
>
> Thanks for your suggestions. We have supplemented relevant results to show that EMoE-learn does benefit from longer fine-tuning to some extent (perhaps every method does; we have not verified it yet), but it still underperforms EMoE. Specifically, we increase training epochs from 20 to 30 in LoRA tuning scenarios. The results (EMoE-learn-long) show some improvement but still underperform EMoE in some tasks (e.g., CoLA). This suggests that while increasing training steps can mitigate slow gate convergence, it might not resolve issues like gate collapse, where training later concentrates on fewer experts (Figure 6c).
> |  | SST2 | RTE | MRPC | CoLA | avg |
> | --- | --- | --- | --- | --- | --- |
> | LoRA | 93.16 | 72.92 | 89.97 | 63.40 | 79.86 |
> | EMoE | 93.50 | 75.21 | 90.85 | 65.33 | **81.22** |
> | EMoE-learn | 93.65 | 71.36 | 89.87 | 64.00 | 79.72 |
> | EMoE-learn-long | 93.73 | 74.49 | 90.07 | 64.13 | 80.61 |
> | EMoE-learn-init | 93.46 | 74.16 | 90.05 | 64.25 | 80.48 |
>
> Additionally, experiments with better initialization (EMoE-learn-init) using the avg-k method for gates showed improvement over EMoE-learn but did not exceed avg-k gating.
>
> > 3. *In GMoEs, auxiliary loss functions (as described in Appendix C.4 of the GMoE paper) are employed to achieve good performance. Were these loss functions used for the GMoE baselines in this study? Additionally, could these loss functions improve the performance of EMoE?*
>
> Yes. The implementation of GMoE used official repo setups, including the loss functions with consistent hyperparameter settings. For EMoE experiments in GLUE and Domainbed, the same MoEs framework and loss functions were used to eliminate interference from factors other than expert construction and gate function. We also highlight this in experiments with the Llama. We achieved significant improvements over the baseline without using any balance loss in our implemented MoEs.
>
> > 4. *The font size of "Office" and "Terra" in the figures appears to be smaller than that of the other labels. It would be helpful to ensure consistency in font sizes across all labels.*
>
> Thanks for your suggestions. We intentionally adjusted the font size because we observed that the character length of "Office" and "Terra" is longer, leading to inconsistent column widths in the formatting and affecting the presentation. We will address this issue.
>
> > 5. *The right figure in Figure 4 seems to lack data points. Could the authors clarify if this is intentional or if there might be an issue with the figure's presentation?*
>
> We notice that the issue occurs when viewing the PDF in a browser but not in a PDF reader. This might be due to the high density of points in the figure. We will try to address this issue in subsequent revisions.

---

> > ### Comment · Reviewer_wdQa · 2023-11-22
> >
> > My concerns have been addressed so I keep the positive score.

---

### Official Review · Reviewer_xYuL · 2023-10-31

**Soundness:** 2 fair
**Presentation:** 3 good
**Contribution:** 1 poor
**Rating:** 3
**Confidence:** 4

**Summary:**

The paper presents a method to leverage the modularity inherent in pretrained backbone models, demonstrating improvements in both in-domain and out-of-domain datasets by inducing modularity in the pretrained backbones. The methodology clusters vectors in lower Feed-Forward Networks (FFN) into multiple experts, activating a few experts per input, which in turn specializes these experts for different types of inputs.

**Strengths:**

1. The proposal to exploit modular structures within pretrained transformers presents a novel approach towards enhancing model performance.
2. The paper provides an interesting analysis illustrating the emergent structure in the FFN layers demonstrating the modular nature of the pretrained backbones.

**Weaknesses:**

1. The improvements noted over Vision Transformer (ViT) and (GMoE) are marginal and fall within the noise range for the provided datasets, as evidenced by the data in Table 8, Appendix 4.1.
2. The method proposed is derived from previous work by Zhang et al. (2022 b), and does not represent an original contribution from the current paper.
3. The paper's method shares similarities with the GMoE method, yet the proposed method achieves comparable performance.
4. Despite the method increasing wall-time by 10% and demanding more memory as stated in Appendix A.3, there is no justification provided to substantiate why the method is beneficial, especially given the negligible performance improvements and the added overhead in hardware implementation.

**Questions:**

1. Could you elaborate on why the EMOE-learn approach did not outperform the avg-k method?

---

> ### Author Response · Authors · 2023-11-17
> **Response to reviewer xYuL (Part 1)**
>
> We first want to thank you for your valuable comments. Please see our responses below to see if they could address your concerns. All explanations will be supplemented in the final version of our paper! More discussions are welcome!
>
> > 1. *The improvements noted over Vision Transformer (ViT) and (GMoE) are marginal and fall within the noise range for the provided datasets, as evidenced by the data in Table 8, Appendix 4.1.*
>
> Thanks for your comments. The different methods (ViT, GMoE, and EMoE) are close to the Domainbed benchmark. However, we want to point out that this is one characteristic of the Domainbed benchmark, owing to its rigorous evaluation setting:  Each dataset comprises 4 distinct domains. One or two domains’ data are sequentially designated within a single training. For example, when training on PACS with four domains {art, cartoons, photos, sketches}, {art, cartoons} could be selected as ID training data, while {photos, sketches} are designated for OOD testing. This configuration results in $C_4^2 + C_4^1 = 10$ training processes within each dataset. Each dataset includes 3 trials. Thus, each result is aggregated from 30 experiments. As a result, they also reported that no algorithm outperformed ViT by a significant margin in their original paper [1]. GMoE achieves SOTA performance, and we achieve comparable performance with it.
>
> Although the performance is not the intended purpose of our paper, we supplement extensive experimental results to support that EMoE brings improvements. Apart from the NLP results in our submission, we supplement experiments using T5-base for multi-task learning and Llama for instruction tuning. Both results indicate that EMoE boosts the ID and OOD performance compared to vanilla fine-tuning. For example, EMoE improves by 7.56 over the baseline on 6 super-GLUE tasks. Please refer to our General Response Table 1 and 2 for complete results.
>
> > *2. The method proposed is derived from previous work by Zhang et al. (2022 b), and does not represent an original contribution from the current paper.*
>
> Our method is indeed derived from [2]. However, we want to highlight that the core contribution of this work is not a method (how to transform a typical Transformer into its MoE counterpart). We want to share the insight and interesting experimental results with the LLM community: **Pre-trained Language Models can benefit from their emergent modularity during the fine-tuning stage.** We call on the entire community to use the emergent modularity feature of the model and to propose more sophisticated methods to externalize the emergent modularity; in our case, as the method is not the first foci of our work, we adopt a simple method from [2].
>
> We highlight that our research question differs from theirs; they have not focused on leveraging the emergent modularity into fine-tuning pre-trained Transformers. The adopted method is one of the variations of their primary method. For further details, please refer to our General Response to all reviewers for discussion between EMoE and MoEfication.
>
> > 3. *The paper's method shares similarities with the GMoE method, yet the proposed method achieves comparable performance.*
>
> As in Question 2, EMoE shares some similarities with GMoE; they both introduce MoE architecture in some layers of the Pre-trained Transformers.
>
> However, we want to highlight the key difference between them. GMoE directly copies the original FFN layer and introduces a learned gate to form an MoE architecture; the different experts of GMoE are initialized identically as the original FFN layer. Therefore, GMoE is inadequate to explore our research question: whether emergent modularity favors fine-tuning transformers. So, we derived our method from MoEfication.  Meanwhile, replicating the FFN layer in the GMoE method makes it less practical for large models (e.g., Llama), but EMoE does not suffer from the same issue. For a detailed comparison, please refer to the table in *Response to reviewer qXUa (Part 1)*.
>
>
> [1] In Search of Lost Domain Generalization Domain Bed, https://arxiv.org/abs/2007.01434
>
> [2] MoEfication: Transformer Feed-forward Layers are Mixtures of Experts, https://arxiv.org/abs/2110.01786

---

> > ### Author Response · Authors · 2023-11-17
> > **Response to reviewer xYuL (Part 2)**
> >
> > > 4. *Despite the method increasing wall-time by 10% and demanding more memory as stated in Appendix A.3, there is no justification provided to substantiate why the method is beneficial, especially given the negligible performance improvements and the added overhead in hardware implementation.*
> >
> > Thank you for pointing this out! We suppose that your concerns derive from the effectiveness and efficiency of EMoE.
> >
> > Regarding efficiency, we found that the increasing wall time and the GPU memory usage come from the public library tutel [https://github.com/microsoft/tutel] that we used to implement EMoE. We reimplemented our method and observed that EMoE does not require significant additional run time and memory usage. Specifically, we introduce an alternative implementation approach in EMoE where hidden states are used to calculate gate scores after computing the first layer. These scores mask the outputs of unselected experts, mimicking the effect of MoEs. Though this theoretically increases FLOPS compared to traditional MoEs, in practice, the speed is comparable to dense models, as demonstrated in the General Response Tables 3, 4, and 5.
> >
> > Regarding effectiveness, we want to argue that EMoE indeed brings some improvements to NLP tasks. Specifically, EMoE achieved an average improvement of 0.84 on 8 GLUE tasks in GPT2-XL without introducing additional parameters. On 6 Super-GLUE tasks in T5-Base, it achieved an average improvement of 6.68. Additionally, on the MMLU benchmark in LLama2-7B, EMoE improved by 1.58. These results demonstrate the effectiveness of EMoE in improving performance across various tasks without significantly increasing model complexity.
> >
> > Furthermore, as mentioned in Section 5.1, EMoE mainly takes effect during training, so one can apply EMoE architecture during training and recompose it into a dense model during inference. EMoE is practical because we do not want our model architecture changed after fine-tuning. And we further validate this idea by instruction-tuning  Llama2-7B and Llama-30B using alpaca. We observe significant improvement in the MMLU benchmark.
> >
> > Given that EMoE does not lead to hardware overhead and is effective and practical, we believe that the EMoE could bring benefits and that externalizing and exploiting the emergent modularity of pre-trained transformers is beneficial.
> >
> > > 5. *Could you elaborate on why the EMOE-learn approach did not outperform the avg-k method?*
> >
> > EMoE with avg-k gating generally performs better than EMoE-learn on average, although there are specific scenarios where EMoE-learn outperforms EMoE with avg-k gating. One possible explanation is that learning gating mechanisms in MoEs can be challenging. EMoE avoids this complexity, allowing it to outperform EMoE-learn in cases where gating doesn't work well. Another potential explanation is that Figure 6 a vs. b suggests that avg-k gating is more stable than learned gating. This stability may help mitigate data inefficiency caused by gating inconsistencies across different training stages, as mentioned in [1].
> >
> > [1] Taming Sparsely Activated Transformer with Stochastic Experts, https://arxiv.org/abs/2110.04260]

---

> > > ### Author Response · Authors · 2023-11-20
> > > **Request for feedback**
> > >
> > > Dear reviewer xYuL
> > >
> > > We have posted responses to clarify the core contribution of EMoE and show its significance and practicality. We wonder if you can let us know whether our responses address your concerns.
> > >
> > > Looking forward to your reply.
> > >
> > > Best regards,
> > >
> > > Authors

---

> > > > ### Comment · Reviewer_xYuL · 2023-11-21
> > > >
> > > > Thank you for your clarifications and additional experiments. It seems to me that this work proposes using the modularity of Feed-Forward Networks (FFNs) in a pretrained backbone to enhance downstream performances. I believe GMoE also offers modularity, even though it wasn't explicitly mentioned in their paper, as the authors have noted. However, the results in the paper are not entirely convincing, comparing to GMoE and regular full-model finetuning. For example, the results in Tables 1 and 2 appear to be within the noise range compared to full model fine-tuning and GMoE.
> > > >
> > > > To better demonstrate the benefits, the authors have undertaken multitask learning on T5-base and also fine-tuned the Llama on Alpaca dataset for MMLU performance, as detailed in the general response. However, there are some issues: 1) The baseline results for T5-Base are lower compared to those in Table 14 of the T5 paper [https://arxiv.org/pdf/1910.10683.pdf](Super GLUE average is 76.2 versus 65.73 reported here). It's possible that the training was not conducted until convergence; if so, I recommend extending the training duration. 2) Fine-tuning LLAMA (both Lora and the full model) on the Alpaca dataset does not seem to benefit the MMLU downstream score at the 7B scale but shows significant improvement at the 30B scale. Could you please double check these results? Even at the 30B scale, the benefits are marginal, within 1 point, and it's unclear if this is significant.
> > > >
> > > > Additionally, the method involves a hyperparameter K, which is challenging to determine initially. This could be a focus for future work by the authors. While the motivations behind this study are clear, I encourage the authors to restructure their work to more convincingly demonstrate the benefits of their method.

---

### Official Review · Reviewer_3WYM · 2023-10-31

**Soundness:** 2 fair
**Presentation:** 3 good
**Contribution:** 2 fair
**Rating:** 3
**Confidence:** 3

**Summary:**

The paper uses a recent method (MoEfication; Zhang, 2022) to convert a pretrained transformer-based model into a mixture-of-experts model and evaluates the transfer performance of the resulting model after fine-tuning on a variety of tasks. It finds improved performance and argues that that this is due to deactivating neurons with negative transfer effects.

**Strengths:**

The paper presentation is clear and the investigated research question is interesting and potentially impactful. The experiments are conducted on a wide range of benchmarks and performance appears to be good.

**Weaknesses:**

The results presented in the tables, spider plots and bar plots in the main text are missing standard errors. Unfortunately this makes it very difficult to judge whether improvements are statistically significant, especially since the means are often very close across methods. I would suggest to add these.

Since the paper uses an existing method, the question of novelity is a bit more subtle. As the authors explain, the main difference of this paper is the focus on understanding transfer performance instead of improving inference efficiency. This puts a stronger weight on the analysis and understanding how EMoE might improve performance. I find some of the evidence presented in the Analysis section not entirely convincing. In particular, I am concerned that the claim "Expert Selection Changes During Training" is not well supported in which case the interpretation of the effectiveness due to modularity (rather than for example pruning) might not be justified. Please see my questions below in this regard.

**Questions:**

1. Given that top-k/N is typically half, I am wondering if this is always the same half of modules activated? Figure6/8 seem to show that some experts are barely selected but this would somehow contradict the claim of the paper that "Expert Selection Changes During Training". Could you please clarify whether I am misunderstanding the figures? Does the fact that a lot of experts are consistently white over the course of training not indicate that they are rarely used in the forward pass?
2. Following up on the previous point, a possible ablation I would have found very informative is to prune the experts based on the selection frequency during training, i.e. fix the gating after training based on the selection frequency for various fractions. I am wondering if the performance gains of EMoE are effectively due to pruning and not due to modularity.
3. I am not sure I understand the ablations conducted to support the claim "EMoE masks neurons with negative transfer impacts.". If we limit the model to use modules with small gating score I don't find it surprising that this performs worth than those with large gating scores as the former might just be mostly zero. Have you checked for this?
4. Following up on the previous point, I would find a more informative ablation to contrast the performance of top-k with selecting all modules i.e. removing the gating entirely. Would it be possible to run such an experiment?

> Differently, we treat these decomposed units as heterogeneous modules to exploit their modular properties in the framework of MoEs, and observe significant improvements over monolithic models.

5. It is not clear to me what it means to "treat decomposed units as heterogeneous modules", could you please clarify this sentence?
6. Why do the hyperparameter grids differ between EmoE and GMoE? How can we exclude the possibility that GMoE potentially also benefits from larger N and/or top-k?
7. I find the point on which layers are selected for the MoEfication puzzling. Why does it make sense to only pick every second layer in the second half of the network?


Typos

- Page 2: "can benefit from its ermgent modular structure"

---

> ### Author Response · Authors · 2023-11-17
> **Response to reviewer 3WYM (Part 1)**
>
> Thanks for your reviews, we hope the following comments could address your concerns.
>
> > 1. *Given that top-k/N is typically half, I am wondering if this is always the same half of modules activated? Figure6/8 seem to show that some experts are barely selected but this would somehow contradict the claim of the paper that "Expert Selection Changes During Training". Could you please clarify whether I am misunderstanding the figures? Does the fact that a lot of experts are consistently white over the course of training not indicate that they are rarely used in the forward pass?*
>
> Thanks for your question. Your understanding of Figure 6 and Figure 8 is correct. For EMoE, some experts are **barely** selected, and the expert choice stabilizes during training. We guess it’s our subheading that causes such misunderstandings. The ‘`changes` ’ here is a noun instead of a verb. We will replace the subheading to avoid the ambiguity.
>
> > 2. *Following up on the previous point, a possible ablation I would have found very informative is to prune the experts based on the selection frequency during training, i.e. fix the gating after training based on the selection frequency for various fractions. **I am wondering if the performance gains of EMoE are effectively due to pruning and not due to modularity.***
>
> Thanks for your suggestions! First of all, regarding your statement, “if the performance gains of EMoE are effectively due to pruning and not due to modularity. ”, we would like to point out that pruning can be regarded as task-level modularity. They cannot be considered as two opposites. Modularity is when specific inputs activate specific modules (or a group of parameters), and pruning is when the useless parameters can be pruned for specific tasks. We can see a strong connection between modularity and pruning; they cannot be decoupled.
>
> Second, we agree that the suggested experiments are informative. Therefore, we supplement two ablations. Their pipelines are:
>
> 1. *Training-Pruning*: Training EMoE ($N$=64, $k$=32), pruning experts with lower selection frequency, and evaluating the pruned model.
> 2. *Pruning-Training*: Pruning lower selection frequency experts of a new model according to 1, then training and evaluating the pruned model. It is important to note that our approach utilizes statistical data from the fine-tuning process to prune a model that had not undergone tuning. This specific configuration is employed to compare pruning and modularity directly during the training phase.
>
> Specifically, we choose BERT-Large within the LoRA tuning framework, where the MoEs component remains untrained. This choice allowed us to gain clear insights into the impact of neurons' selection within the FFNs on Lora weight learning, consistent with the settings employed in Analysis 5.1.
>
> The experimental results are presented in the table below. Our findings provide further substantiation for our explanation of EMoE, highlighting that EMoE's effects manifest during the fine-tuning stage rather than during inference, as also underscored in our response to Question 5. Our main observations are:
> 1. When we prune sparsely selected experts during inference ($N$=64, $k$=16,32 scenarios), we observed comparable performance with EMoE. This observation held true across various scenarios, consistently surpassing the baseline.
> 2. When we prune some barely selected experts according to training statistics from the first ablation and fine-tune the remaining experts, the performance gains attributed to EMoE become imperceptible. Even in the most optimal setting ($k$=32, 63.52), the performance only marginally exceeded the baseline (63.40) and notably fell short of the results achieved by training with EMoE, followed by pruning to $k$=32 (65.33).
>
>     In conclusion, we argue that the pruning and modularity are homologous to some extent. They cannot be regarded as opposites. And further experiments support our explanation in the submission. And we will add these additional ablation results in the final version of our work.
>
>     | |  CoLA |  |  | SST2 |  |
>     | --- | --- | --- | --- | --- | --- |
>     | LoRA | 63.40 |  |  | 93.16 |  |
>     | EMoE (k=32, N=64) | 65.33 |  |  | 93.54 |  |
>     | |  |  |   | |  |
>     | Expert number after pruning | Training-Pruning | Pruning-Training |  | Training-Pruning | Pruning-Training |
>     | 64 | 65.26 | 63.4  |  | 93.42 | 93.16 |
>     | 32 | 65.33 | 63.52 |  | 93.54 | 93.34 |
>     | 16 | 65.33 | 63.42 |  | 93.45 | 93.27 |
>     | 8 | 64.33 | 63.42 |  | 93.45 | 93.27 |
>     | 4 | 63.80 | 63.21 |  | 93.34 | 93.21 |
>     | 1 | 63.42 | 62.42 |  | 93.34 | 92.58 |

---

> > ### Author Response · Authors · 2023-11-17
> > **Response to reviewer 3WYM (Part 2)**
> >
> > > 3. *I am not sure I understand the ablations conducted to support the claim "EMoE masks neurons with negative transfer impacts.". If we limit the model to use modules with small gating score I don't find it surprising that this performs worth than those with large gating scores as the former might just be mostly zero. Have you checked for this?*
> >
> > In the case of using avg-k gating in EMoE, we are using a hard gate. The gate scores are either 0 or 1 (Section 3.2). Therefore, when restricting the model to use only modules with small gating scores, the outputs of these experts are still weighted as 1, and there is no influence from gating scores being 0.
> >
> > However, we think the scenario related to the reviewer's concern is when the selected experts contain mostly inactive neurons that do not contribute to the model's performance.  Therefore, we have also followed the reviewer's suggestions and conducted experiments to examine the activation patterns of neurons in different experts concerning input tokens. This allows us to investigate whether the experts with small gating scores contain no activated neurons.
> >
> > In the table, the activation ratio is determined by calculating the proportion of activated neurons that belong to the selected expert among all activated neurons. Meanwhile, the weighted activation ratio is computed by taking the ratio of the sum of activation scores in the selected experts to the sum of the activation scores across the entire FFN layer.
> >
> > | SST2 |  | Top selection |  | Bottom selection |  |
> > | --- | --- | --- | --- | --- | --- |
> > | activation ratio | k | EMoE | EMoE-learn | EMoE | EMoE-learn |
> > |  | 32 | 0.6879 | 0.6796 | 0.3121 | 0.0552 |
> > |  | 16 | 0.4317 | 0.4247 | 0.124 | 0.1293 |
> > |  | 8 | 0.2616 | 0.2558 | 0.0528 | 0.3218 |
> > | weighted activation ratio |  |  |  |  |  |
> > |  | 32 | 0.731 | 0.7234 | 0.269 | 0.2791 |
> > |  | 16 | 0.4953 | 0.4888 | 0.1041 | 0.109 |
> > |  | 8 | 0.3304 | 0.3237 | 0.0442 | 0.046 |
> > |  |  |  |  |  |  |
> > | CoLA |  |  |  |  |  |
> > | activation ratio |  | EMoE | EMoE-learn | EMoE | EMoE-learn |
> > |  | 32 | 0.6815 | 0.6788 | 0.3185 | 0.3212 |
> > |  | 16 | 0.4292 | 0.4239 | 0.1295 | 0.131 |
> > |  | 8 | 0.264 | 0.2582 | 0.0552 | 0.0637 |
> > | weighted activation ratio |  |  |  |  |  |
> > |  | 32 | 0.7341 | 0.4958 | 0.2659 | 0.2744 |
> > |  | 16 | 0.512 | 0.3389 | 0.0468 | 0.1109 |
> > |  | 8 | 0.3589 | 0.7256 | 0.1063 | 0.0488 |
> >
> > It can be observed that:
> >
> > 1. Even when selecting the bottom 1/8 in EMoE, the activation ratios are still around 0.1, not entirely 0. In the case of random selection, the activation values should be around 0.125. We also observed that the bottom-k selection still results in fewer inactive neurons than random-k. This indicates that the negative transfer we mentioned is not solely due to selecting entirely inactive neurons.
> > 2. In EMoE, both top-k and bottom-k gating perform better than learned gating in selecting neurons with a higher activation ratio and output weight. This suggests that avg-k gating is more effective than learned gating in helping the model choose neurons.
> >
> > In addition to these observations, our supplementary experiments on T5 also found that EMoE exhibits a more significant improvement relative to the baseline in multi-task learning settings, which aligns with the idea of mitigating negative transfer.
> >
> >
> > > 4. *Following up on the previous point, I would find a more informative ablation to contrast the performance of top-k with selecting all modules i.e. removing the gating entirely. Would it be possible to run such an experiment?*
> >
> > Thanks for your suggestions. However, it seems that there are some misunderstandings between us. If we remove the gating and fine-tune the model, this is our dense fine-tuning baseline. If we first fine-tune the EMoE model and then remove the gating for inference. We point out that this is related to our experiments and analysis in Section 5.1, where we concluded that "**EMoE benefits LoRA weights learning instead of influencing inference**." The EMoE2LoRA experiments involve removing gating after training EMoE and using all the experts for inference. As shown in Figure 3 (left), the inference results with selecting all modules in the LoRA tuning scenario are comparable to those of EMoE. Additionally, in the supplementary experiments related to LLama, we used all modules during inference in both full-finetuning and LoRA tuning settings. We observed significant improvements relative to the baseline, which further supports the analysis results in our paper.
> >
> > We look forward to further discussions that may help us address your concern on this point.

---

> > > ### Author Response · Authors · 2023-11-17
> > > **Response to reviewer 3WYM (Part 3)**
> > >
> > > > 5. *It is not clear to me what it means to "treat decomposed units as heterogeneous modules", could you please clarify this sentence?*
> > >
> > > The sentence here highlights the difference between the motivation behind EMoE and MoEfication. We realize that this statement potentially causes clarity issues in our work, and we will rectify it in our final version. For now, please see our explanations below.
> > >
> > > 1. MoEfication mainly seeks to improve inference efficiency. They decompose the FFNs into sparse MoE **after fine-tuning** so that sparsely activated experts can approximate the functionality of the original FFNs with reduced computational cost.
> > > 2. Differently, EMoE wants to externalize the emergent modular nature of the pre-trained transformers so that the different experts are sparsely updated and are further encouraged to achieve distinct specializations. So, we decompose FFNs into sparse MoEs **before fine-tuning** on downstream tasks. Though similar methods are applied, EMoE provides a different view about what we can benefit from emergent modularity, and this is what we want to share with the community with this work.
> > > 3. When we mention "heterogeneous," we compare EMoE to other methods that introduce experts (e.g., GMoE and Upcycling[https://arxiv.org/abs/2212.05055]). EMoE's experts are directly derived from pre-trained FFNs, and this origin makes them more diverse or "heterogeneous" in terms of their functionality compared to other approaches like GMoE or upcycling. These other methods may not exhibit as much diversity among their experts because their experts are initialized by directly copying existing FFNs.
> > >
> > > > 6. *Why do the hyperparameter grids differ between EMoE and GMoE? How can we exclude the possibility that GMoE potentially also benefits from larger $N$ and/or top-k?*
> > >
> > > First, One key underlying reason is that EMoE decomposes the original FFNs into experts; no matter how we increase $N$ and $k$, the total parameters of all experts will remain the same as the original FFNs. On the contrary, GMoE replicates the original FFN from a MoEs architecture; the larger $N$ and top-k they choose, the more parameters and computational overhead they introduce. To put it simply, if there are $h$ neurons in the FFNs:
> > >
> > > 1. EMoE uses (top-k/$N$) $\times h$  neurons, which is a fraction of the total number of neurons in FFNs
> > > 2. GMoE uses top-k $\times h$  neurons, which is a larger number of neurons compared to EMoE
> > >
> > > Second, the hyper-parameters of GMoE are mainly from their original paper. The settings of MoEfication influence the choice of $N$ and top-k in EMoE. The primary principles behind these choices are as follows:
> > >
> > > 1. The size of each expert should be moderate. If experts are too large, it can lead to the compression of information in the keys during the avg-k gating process.
> > > 2. When using top-k activation, the number of neurons selected should be 1/4 to 1/8 of the total number of neurons in the FFNs. Choosing too few neurons can deviate significantly from the original functionality of the FFNs, while selecting too many would eliminate the desired sparsity.
> > >
> > > At last, we verify that GMoE will not benefit from larger $N$ and $k$ by supplementary experiments. We conduct LoRA tuning (because a very large $N$ of GMoE introduces too many additional parameters, we only tune the Lora module and the layer where GMoE introduces experts to test GMoE on very large $N$) on the GLUE benchmark; we examined the impact of increasing $N$ and top-k in GMoE, as shown in the table below:
> > >
> > > |   |   |   |   |   |   |
> > > |---|---|---|---|---|---:|
> > > |Method|N|Top-k|CoLA|SST2|trainable parameters / original parameters|
> > > |GMoE|4|1|63.87±0.57|93.88±0.11|0.28|
> > > ||8|2|63.80±0.71|93.85±0.11|0.54|
> > > ||16|2|64.15±0.52|93.58±0.21|0.89|
> > > ||16|4|63.80±0.53|93.61±0.19|0.89|
> > > ||16|8|63.27±0.31|93.69±0.16|0.89|
> > > ||64|32|63.27±0.42|93.81±0.23|3.29|
> > > |LoRA|||63.40±0.62|93.16±0.19|000.2|
> > > |EMoE|64|32|65.33±0.40|93.50±0.33|000.2|
> > >
> > > It can be observed that EMoE achieves a significant improvement over LoRA without introducing additional parameters relative to LoRA. On the other hand, GMoE's performance does not necessarily improve as $N$ and top-k increase, and it comes at the cost of significantly increasing the number of parameters. This highlights the efficiency and effectiveness of EMoE compared to GMoE.
> > >
> > > > *7. I find the point on which layers are selected for the MoEfication puzzling. Why does it make sense to only pick every second layer in the second half of the network?*
> > >
> > > Our approach's "every second layer" choice follows the configuration established in GMoE. We found that having too many MoE layers can lead to a degradation in results. The decision to focus on the "second half" of the network is based on prior research findings, specifically from the work [https://openreview.net/forum?id=TJ2nxciYCk-] This research indicates that deeper layers in neural networks tend to exhibit more pronounced sparsity.

---

> > > > ### Author Response · Authors · 2023-11-20
> > > > **Request for feedback**
> > > >
> > > > Dear reviewer 3WYM
> > > >
> > > > We have posted extra experiment results in general responses and point-wise answers to your questions. We wonder if you can let us know whether our responses address your concerns.
> > > >
> > > > Looking forward to your reply.
> > > >
> > > > Best regards,
> > > >
> > > > Authors

---

> > > > > ### Comment · Reviewer_3WYM · 2023-11-21
> > > > > **Response to rebuttal**
> > > > >
> > > > > First and foremost I would like to thank the authors for their thorough response to my questions and suggestions. I very much appreciate the effort spent on additional ablations and experiments to substantiate their response. I apologize that it took me so long to give feedback.
> > > > >
> > > > > You response has clarified most of my questions. Unfortunately, I still find it difficult to recommend the current version of the paper for acceptance.
> > > > > As I stated in my initial review, "Since the paper uses an existing method, the question of novelty is a bit more subtle. [...] This puts a stronger weight on the analysis and understanding how EMoE might improve performance". Given the findings on the topic of pruning/task-level modularity, I believe the current draft does not sufficiently address this point:
> > > > >
> > > > > The authors seem to agree with me on the point that the success of EMoE is likely associated to task-level pruning rather than for instance example-specific modularity.
> > > > > I also agree with the authors' view that these are related phenomena and pruning can be seen as task-level modularity where the activation of modules is fixed during inference. However, the way the current draft motivates and explains the experiments and results is heavily based on the narrative of "emergent modularity". In my opinion this gives a false impression of why the suggested fine-tuning procedure is effective.
> > > > > If the method's success is due to pruning, experiments should be designed to compare it to other pruning methods when fine-tuning pretrained networks.
> > > > > Even when maintaining the perspective of task-level modularity, the method should be clearly presented in this light and the experimental analysis could for instance demonstrate that indeed different "modules" are selected for different tasks.
> > > > >
> > > > > That being said I would like to encourage the authors to revise their analyses in light of the interesting experimental results obtained already to better substantiate why this method works for future submissions.
> > > > >
> > > > > One additional point of my initial review has not been addressed that I would like to emphasize again. It is not the main reason for why I am not recommending this paper for acceptance but I nevertheless think it should be fixed in future submissions:
> > > > >
> > > > > > The results presented in the tables, spider plots and bar plots in the main text are missing standard errors. Unfortunately this makes it very difficult to judge whether improvements are statistically significant, especially since the means are often very close across methods.

---

### Official Review · Reviewer_6Xtr · 2023-11-02

**Soundness:** 4 excellent
**Presentation:** 3 good
**Contribution:** 3 good
**Rating:** 8
**Confidence:** 3

**Summary:**

This work proposes Emergent mixture-of-Experts, for constructing a modular architecture from a pre-trained transformer (post pre-training), without introducing any extra parameters to the model. The focus of this paper is on measuring generalization performance and if the proposed method can provide improved generalization in different settings. This work aims to show the generalization benefits that can be obtained by modularity that “emerges” after pre-training, rather than the modularity that is “baked-in” a model from the start or after pre-training (as done by other methods).

**Strengths:**

1. Strong empirical study: experiments are sufficient and conducted over different modalities and model architectures, ablations are sufficient and extensive as well.
2. Clarity: This work has made clear their contributions, and the paper is easy to understand and follow.
3. Interesting results: EMoE is shown to match or improve the performance of GMOE which requires more parameters for its construction, this result shows that you don't really need to have those extra parameters. Results are presented across DomainBed, and GLUE and over fine-tuning and lora-tuning settings. EMoE also improves over LORA which is an interesting result.
4. Analysis: The analysis part of the paper is also quite interesting. Showing that EMoE masks neurons which cause negative transfer of knowledge, I have some doubts on the other analysis which is done in the paper (see Q 1. below).

**Weaknesses:**

There are no major weaknesses that I see in this study. The work is sound in terms of experimentation and through various results and analysis is able to sufficiently address the main question of the paper as to whether EMOEs are beneficial for downstream tasks from the generalization perspective. There are some limitations as described in Section 6, that we are not sure how much the results hold for larger LLMs and for more involved tasks. As made clear by the paper, this work does not provide any new methods (for MOE construction), however, the analysis and results give a lot of insight and hence I believe novelty is not an issue with this paper.

**Questions:**

1. I am not sure if I understand the first analysis of Sec. 5. Completely. Are the conclusions of this analysis as follows: a) since LoRA2EMoE does not lead to much difference with respect to LoRA, hence sparse activation does not have much impact during testing. And b) Since EMoE2LoRA matches EMoE and leads to performance gains over LoRA, we can say that having modularity actually helps while fine-tuning. If my understanding is correct, I feel this section can be better written with more explanation in simple terms. If not, then I would like to know from authors a more detailed explanation and flaws in my understanding.

---

> ### Author Response · Authors · 2023-11-17
> **Response to reviewer 6Xtr**
>
> Thanks for your suggestions. Your understanding of the analysis in Section 5 is entirely right. We want to emphasize here that EMoE mostly influences the training stage instead of inference. And we also highlight that we can use original architecture for inference after EMoE-style fine-tuning. This makes our method more practical (because we do not need to change the model architecture after fine-tuning) and potentially contributes more to the community. We will re-write this part to make it easier to follow.
>
> Regarding our limitations, we have addressed them in *Section 6: Can larger LLMs benefit from its emergent modularity?* Specifically, we employ the Alpaca dataset to instruction-tune Llama2-7B Llama-30B models in the EMoE manner and use MMLU as the evaluation benchmark. We observe consistent improvement in MMLU performance compared with vanilla fine-tuning. Please refer to Tables 3, 4, and 5 in General Response for full results. I hope such additional results can address our minor limitations and support you in recommending our paper!

---

### Official Review · Reviewer_qXUa · 2023-11-07

**Soundness:** 2 fair
**Presentation:** 2 fair
**Contribution:** 2 fair
**Rating:** 3
**Confidence:** 4

**Summary:**

This work introduces a new neural network layer which can immediately replace the dense layers within transformer modules. To train the layers first a dense layer is pre-trained. The incoming weight matrix to the hidden layer is then clustered into $N$ centroids such that all hidden neurons with the same features are grouped in a module. The average feature is then used as a key to identify when this module should be used for inference. At inference the $N$ modules keys are compared to the input (the query) and the top $k$ modules are used to forward propagate the input. The outputs of these modules are then used in a weighted sum where the weighting is determined by a gating function. A hand-crafted gating function as well as learned gatings are explored empirically but the hand-crafted function is found to be superior. A number of datasets are experimented on with proposed Emergent Mixture of Experts model (EMoE) and an ablation is also conducted to determine the utility of each part of the layer's computation.

**Strengths:**

## Originality
This work is draws heavily on prior work, particularly GMoEs and MoEfication, but remains fairly original. It is clearly stated what the intended differences are from prior work. For GMoEs this is the absence of additional parameters or training. In the case of MoEfication the decomposed units are treated as "heterogeneous modules" which avoid the drop in performance seen from MoEfication.

## Quality
The experiments employed in this work are extensive and span a number of practical domains, such as vision and text based tasks. Challenging datasets are also used and the algorithm is compared to a number of challenging baselines. In addition the ablation study is interesting and helpful for understanding the various components of the EMoE architecture. Overall the evaluation of EMoE aims to be thorough and indeed challenges EMoE.

## Clarity
The figures are a particular high-point of this work, especially Figure 1 which is quite helpful for understanding the proposed EMoE layer, while Figure 2 is a neat and visually intuitive summary of the results.

## Significance
The proposed layer is indeed significance and would provide a clear contribution to the field. Specifically, a mechanism to zero-shot identify modules within a dense network that improves the generalizability of the overall network and are more interpretable. I could also see future work continuing to build on the proposed layer and introduce more sophisticated ways of identifying modules or calculating the gating function for example. The ablation study will indeed help with guiding this future work.

**Weaknesses:**

## Quality
My primary concern of this work lies in the interpretation of the results and the intended message which they aim to convey. The authors claim that there is a benefit from EMoE and I cannot see how this is consistently the case across Tables 1, 2 and 3. While there is some improvement over the various baselines in each case (ViT, BERT, GPT2 and LoRA) it still appears marginal and fairly inconsistent. This is  made worse by the fact that standard deviations are not reported or commented on (while they are referenced in the caption of Table 4.1). On top of this, there is even less benefit when comparing to GMoE, however the authors mention in the discussion that being competitive towards GMoE is not the intended purpose. However, it is still stated that "Overall, EMoE outperforms ViT and GMoE". Which then highlights my second concern on quality. It is unclear to me what the intended purpose is of this work. It is stated in the introduction that the work aims to "explore whether and how standard pre-trained transformers could benefit from emergent modular structures". But this question seems to have been answered by the two prior works. GMoEs would indicate that there is a generalization benefit, and MoEfication demonstrates that there is a computational benefit. At stages it seems that EMoEs are being proposed because they do not add parameters or require more training like GMoEs, which I agree would be important. However, at no point is this said and at no point is an substantive evidence given to demonstrate a computational benefit for EMoEs. So based on the experiments this does not seem to be the case. A lot of my confusion here may be secondary due to the issues with clarity which I will now outline. Finally, while I appreciate the long ablation and it is indeed a helpful part of this work, determining the number of centroids in the clustering and number of modules to use are two key hyper-parameters which are not considered or spoken about at all. These two parameters can determine entirely how the module generalizes or under-fits and its overall expressive power. They are also two parameters which must now be tune -- together. Is this addition of complexity worth the benefit over using the vanilla baselines? The answer to that question is likely up to the reader, but failing to mention this or provide the reader with evidence on this appears to be a significant omission. I appreciate that Figure 5 does touch on the expert selection fraction but this is not exactly what I am suggesting and this experiment only uses EMoE architectures and compares to a dense network (it is part of the ablation so I understand that in this context limiting to these comparisons make sense, but still this leaves a gap in the experimentation and paper as a whole).

## Clarity
Firstly, in many cases this work is vague. Even in the abstract using the phrase "...superior out-of-generalization, learning efficiency, etc." is strange, particularly due to the use of "etc" with little context with which to determine what is being referred to. A second example, and the most severe in this work is it is the statement which describes how EMoEs differ from MoEfication is "we treat these decomposed units as heterogeneous modules to exploit their modular properties in the framework of MoEs". What is a "heterogeneous module" in this context. Surely these modules are heterogeneous by some metric by definition since they are identified by a clustering algorithm? But then why would MoEfication have homogeneous modules? A final example, which is similarly important for understanding how EMoEs work says "EMoE and dense model only differ in FFNs activations". Surely the EMoE differs by the fact that it is now gated with an enforced sparsity?

A second note on clarity, I relied on Figure 1 to understand EMoEs as overall Section 3.1 is difficult to parse. Having the transpose on the weight matrix $K^T$ just makes it difficult to follow. Additionally, the output of the $x \cdot K^T$ computation is a $d$-dimensional vector but you try multiply it on the right by a $h \times d$ matrix $V$. So $V$ must also be transposed here. Further, vectors are usually assumed to be column vectors when they are being operated on by matrices and so to following the matrix multiplication of $x$ you have to implicitely transpose it. This is a lot to try and keep in memory while trying to ground the work with the proposed network layer. As a minor extension of this, when would $d$ represent the dimension of the hidden layer and $h$ represent the dimension of the data? Also the first usage of $K_i$ does not bold the $i$ but it is bold everywhere else, so that should be corrected (that's not a critique, just wanted to point it out). Finally, while I appreciate that this was aiming to be grounded within the transformer literature , rephrasing a normal neural network forward pass in terms of keys, queries and values is also just unhelpful. Especially when the work is inconsistent on what the key is. Section 3.1 says the keys are the rows of $K$ but then it is more accurate to interpret the centroids of the clusters of $K$ as the keys as this is what is being compared to the query (input) to determine which module to use.

Finally, the other architectures and algorithms which are compared against are not introduced sufficiently. This makes it very difficult to assess the contributions of EMoE and its practical applicability. Moreover it makes it difficult to interpret the results. For example, why are EMoEs given significantly more modules to train (a larger $N$) and use (a larger $k$) than GMoEs? Does this make a comparison unfair? I also feel that the main take-away of the paper: "Instead, the goal is to demonstrate that dense pre-trained transformers can benefit from emergent modular structures" does not quite summarize accurately what is shown. This would demonstrate that the emergent modular structure is a phenomenon which lasts in transformers, not some structure which must be algorithmically extracted and enforced. But this is a final minor critique on clarity.

Overall I would be inclined to increase my score. Particularly if the results incorporate standard deviations and the discussion is updated to more accurately analyze the results. Unfortunately it seems EMoE does not merely out-perform the baselines. Secondly, I would also substantially increase my score if the clarity concerns are addressed above. I recommend cutting down on Sections 3.1 and 3.2 as they are not particularly helpful and Figure 1 does a great job expressing the same thing. I also think Section 2 could be shortened for a more in-depth background section focusing on GMoE and LoRA and potentially MoEfication which focuses on how EMoE differs. Finally, a general check of the grammar and vague phrasing is in order. While I believe we should be sympathetic towards use of language in these reviews, in this case it makes understanding the paper very difficult. I am also open to revising my score if it is shown that I have indeed missed a crucial point of this work or its presentation.

**Questions:**

I have asked a number of questions in the Weaknesses section above where they came up naturally within the context of my review. I would appreciate if these questions were covered. I do not believe I have any further questions for this section at this time.

---

> ### Author Response · Authors · 2023-11-17
> **Response to reviewer qXUa (Part 1)**
>
> Thanks for your valuable comments on the quality and clarity of our work. We agree that some statements in our submission may be unclear, and we will revise our submission to make it easier to follow (We will revise it by the end of our discussion period). Before that, we want to give more explanations of our work because we think there might be some misunderstandings in the reviews.
>
> **First**, we would like to respond to your primary concern: **what message do we want to convey?**  We want to share with the reader that pre-trained transformers exhibit emergent modularities (different groups of parameters may spontaneously specialize in different functions), and such emergent modularity should be further encouraged at the training stage of the model to achieve the benefit of ID and OOD performance. We verify our point of view with extensive experiments. It turns out that EMoE generally outperforms vanilla fine-tuning.
>
> We understand your concern that some improvements are marginal, such as compared to ViT on Domainbed (For detailed reasons, please refer to our response to reviewer xyul question 1). We thus supplement stronger experimental evidence to address your concern. When fine-tuning T5-base models on  6 Super-GLUE tasks, EMoE achieved an average improvement of 6.68%. Additionally, when instruction-tuning Llama2-7B using the Alpaca dataset, EMoE significantly improved the score on MMLU benchmark by 1.58. These results demonstrate the effectiveness of EMoE. Please refer to the general response Tables 3,4 and 5 for complete results. Given such stronger evidence, we believe there is a benefit of EMoE.
>
>
> **Second**, as the relationship between EMoE, GMoE, and MoEfication has been repeatedly questioned, we want to give more explanations.
>
> Having observed that FFNs layers in pre-trained transformers are sparsely activated (many neurons are unused for inputs), **MoEfication** splits transforms FFNs into a sparse MoEs, aiming to approximate the functionality of the original FFNs to reduce the computational cost, further improving inference efficiency.  Besides, the **GMoE** makes multiple replicates of the original FFNs layer and introduces a learned gate to form a MoEs architecture. They claim that such architecture could improve OOD performance from their theoretical perspective of algorithmic alignment framework [https://openreview.net/forum?id=rJxbJeHFPS]. MoEfication and GMoE do not touch on how emergent modularity influences the training stage. The table below illustrates a detailed comparison of these works.
>
> A comprehensive comparison between EMoE, MoEfication, and GMoE:
>
> |   |   |   |   |
> |---|---|---|---|
> |Aspect|EMoE|MoEfication|GMoE|
> |Research Problem|Exploit the emergent modularity during fine-tuning pre-trained transformers|Approximate FFNs with sparse MoEs to improve inference efficiency.|Validate the OOD improvements brought by Sparse MoEs architectures.|
> |Methods|Splits of FFNs|Splits of FFNs|Copies of FFNs|
> |Practicality|No additional trainable parameters. Experts can be recomposed into the dense model so that models can be deployed as a standard model.|May need re-training on the original task. May suffer from inference latency owing to the specific implementation of MoEs architectures.|Additional trainable parameters are introduced. May suffer from inference latency owing to the specific implementation of MoEs architecture.|
> |Contribution|Significant general improvement without adding parameters|Improved inference efficiency (depending on the specific implementation of MoEs), but performance drop|Significant OOD improvement with additional parameters|
>
> We thank you for your valuable feedback regarding clarity. We apologize for our messed notation. We will explain your questions in your reviews below and revise our submission by the end of the Author-Discussion period (We still need some time to work on it). Please first see if our explanation could address your concerns.

---

> ### Author Response · Authors · 2023-11-17
> **Response to reviewer qXUa (Part 2)**
>
> Please see our point-wise responses to the question in your review.
>
> > 1. *The authors claim that there is a benefit from EMoE and I cannot see how this is consistently the case across Tables 1, 2 and 3. While there is some improvement over the various baselines in each case (ViT, BERT, GPT2 and LoRA) it still appears marginal and fairly inconsistent. This is made worse by the fact that standard deviations are not reported or commented on (while they are referenced in the caption of Table 4.1)*
>
> To clarify the presentation of our results, we have included the results with standard deviations in the appendix, specifically in Tables 8, 9, and 10. We acknowledge the reviewer's concern regarding the significance of our results; therefore, we have provided additional and more robust experimental evidence. In the context of the Super-GLUE tasks within the T5-Base framework, our model demonstrated a noteworthy average improvement of 6.68%. Furthermore, when evaluating the Alpca-tuned Llama2-7B on the MMLU benchmark, EMoE exhibited a substantial improvement of 1.58. In the case of the Llama-30B dataset, EMoE also demonstrated a notable improvement of 0.93.
>
> > 2. *"explore whether and how standard pre-trained transformers could benefit from emergent modular structures". But this question seems to have been answered by the two prior works. GMoEs would indicate that there is a generalization benefit, and MoEfication demonstrates that there is a computational benefit.*
>
> **We want to emphasize that neither GMoEs nor MoEfication has addressed our research question.** We will provide a more detailed introduction to GMoEs and MoEfication, followed by a comparison highlighting the distinctions between these approaches and EMoE.
>
> 1. MoEfication does not delve into the properties of these split experts or explore alternative ways to utilize them beyond enhancing efficiency. MoEfication is motivated by the observation that the activation of neurons within the pretrained transformer FFNs exhibits sparsity and certain cooperative characteristics. Specifically, within the two matrix operations in FFNs, most of the computations do not impact the output. Therefore, MoEfication aims to split the matrices into the form of experts, allowing input data to be involved only in computations performed by experts that influence the output. This involves using sparsely activated experts to approximate the functionality of the original FFNs, leading to reduced computational costs. **Importantly, all these operations are conducted after fine-tuning, and MoEfication's primary objective is to make FFNs computations more efficient.**
>
> 2. GMoE doesn't capture the modular structures within the FFNs of pretrained transformers. GMoE is developed based on theoretical insights, recognizing that MoE architectures contribute to models' out-of-distribution (OOD) generalization. In GMoE, pretrained transformer FFNs are duplicated to construct MoE architectures during the tuning phase, with validation conducted in OOD settings. It's important to highlight that the experts in GMoE are initialized identically to the original FFNs layer and do not delve into the inner structure of the FFNs layer. **As a result, GMoE is inadequate to explore our research question: whether externalized emergent modularity favors fine-tuning of transformers.**

---

> ### Author Response · Authors · 2023-11-17
> **Response to reviewer qXUa (Part 3)**
>
> > 3. *determining the number of centroids in the clustering and number of modules to use are two key hyper-parameters which are not considered or spoken about at all. These two parameters can determine entirely how the module generalizes or under-fits and its overall expressive power. They are also two parameters which must now be tune -- together. Is this addition of complexity worth the benefit over using the vanilla baselines? The answer to that question is likely up to the reader, but failing to mention this or provide the reader with evidence on this appears to be a significant omission.*
>
> Our focus is primarily on investigating the effectiveness of emergent modularity in downstream tasks. We have adopted the MoEfication's partitioning method directly, so we did not delve extensively into the settings of relevant hyperparameters. We largely followed the conventions established by MoEfication. As for the settings of $k$ and $N$, there is further discussion in our response to question 6.
>
> We believe that EMoE is not highly sensitive to the hyperparameters $N$ and $k$; thus, **it does not introduce significant additional complexity.** Our supplementary experiments show that EMoE consistently outperforms the baseline on the new T5 multi-tasks and LLama datasets when using the empirical choice of fractions, 1/4 or 1/8, identified on the GLUE benchmark.
>
> Moreover, we want to highlight the effectiveness and practicality of EMoE. In the Llama experiments, besides no additional parameters, EMoE doesn’t introduce a significant increase in wall time and FLOPS compared with the baselines. However, EMoE-tuned LLama2-7B improves by 1.58 on the MMLU benchmark. Furthermore, as mentioned in Section 5.1, EMoE mainly takes effect during training, so one can apply EMoE architecture and recompose it into a dense model during inference. This property ensures that the EMoE-trained model can be deployed without changes in model architecture— a characteristic neither GMoE nor MoEfication can achieve.
>
>
> > 4. *we treat these decomposed units as heterogeneous modules to exploit their modular properties in the framework of MoEs". What is a "heterogeneous module" in this context. Surely these modules are heterogeneous by some metric by definition since they are identified by a clustering algorithm? But then why would MoEfication have homogeneous modules?*
>
> The sentence here highlights the difference between the motivation behind EMoE and MoEfication. We realize that this statement potentially causes clarity issues in our work, and we will rectify it in our final version. For now, please see our explanations below.
>
> 1. MoEfication mainly seeks to improve inference efficiency. They decompose the Feed-Forward Networks (FFNs) into sparse MoE **after fine-tuning** so that sparsely activated experts can approximate the functionality of the original FFNs with reduced computational cost.
> 2. Differently, EMoE wants to externalize the emergent modular nature of the pre-trained transformers so that the different experts are sparsely updated and are further encouraged to achieve distinct specializations. So, we decompose FFN into sparse MoE **before fine-tuning** on downstream tasks. Though similar methods are applied, EMoE provides a different view about what we can benefit from emergent modularity, and this is what we want to share with the community with this work.
> 3. When we mention "heterogeneous," we compare EMoE to other methods introducing experts (e.g., GMoE and upcycling). EMoE's experts are directly derived from pre-trained FFNs, and this origin makes them more diverse or "heterogeneous" in terms of their functionality compared to other approaches like GMoE or upcycling. These other methods may not exhibit as much diversity among their experts because their experts are initialized by directly copying existing FFNs.
>
> > 5.  *Even in the abstract using the phrase "...superior out-of-generalization, learning efficiency, etc." is strange, particularly due to the use of "etc" with little context with which to determine what is being referred to.*
>
> We appreciate the feedback provided by the reviewer, and we will remove the phrase and refine the language in our revision.

---

> ### Author Response · Authors · 2023-11-17
> **Response to reviewer qXUa (Part 4)**
>
> > 6. *why are EMoEs given significantly more modules to train (a larger $N$) and use (a larger $K$) than GMoEs? Does this make a comparison unfair?*
>
> The choice of different $N$ and $k$ settings is associated with the sources of experts in EMoE and GMoE. We will explain the rationale and impact of selecting $N$ and $k$ in EMoE and GMoE. Regardless of how $N$ and $k$ are chosen, the training parameters of GMoE are significantly greater in number compared to EMoE. Therefore, from the perspective of training parameters, the comparison is fair. In addition, our supplementary experiments demonstrate that this comparison does not affect our conclusions regarding the effectiveness of EMoE.
>
> First, One key underlying reason is that EMoE decomposes the original FFNs into experts; no matter how we increase $N$ and $K$, the total parameters of all experts will remain the same as the original FFNs. On the contrary, GMoE replicates the original FFN from a MoEs architecture; the larger $N$ and top-k they choose, the more parameters and computational overhead they introduce. To put it simply, if there are h neurons in the FFNs:
>
> 1. EMoE uses $h$ * (top-k/$N$) neurons, which is a fraction of the total number of neurons in FFNs
> 2. GMoE uses $h$ * top-k neurons, which is a larger number of neurons compared to EMoE
>
> Second, we verify that GMoE will not benefit from larger $N$ and $k$ by supplementary experiments. We conduct LoRA-tuning (because a very large $N$ of GMoE introduces too many additional parameters, we only tune the Lora module and the layer where GMoE introduces experts to test GMoE on very large $N$) on the GLUE benchmark; we examined the impact of increasing $N$ and top-k in GMoE, as shown in the table below:
>
> |   |   |   |   |   |   |
> |---|---|---|---|---|---:|
> |Method|N|Top-k|CoLA|SST2|trainable parameters / original parameters|
> |GMoE|4|1|63.87±0.57|93.88±0.11|0.28|
> ||8|2|63.80±0.71|93.85±0.11|0.54|
> ||16|2|64.15±0.52|93.58±0.21|0.89|
> ||16|4|63.80±0.53|93.61±0.19|0.89|
> ||16|8|63.27±0.31|93.69±0.16|0.89|
> ||64|32|63.27±0.42|93.81±0.23|3.29|
> |LoRA|||63.40±0.62|93.16±0.19|000.2|
> |EMoE|64|32|65.33±0.40|93.50±0.33|000.2|
>
> It can be observed that EMoE achieves a significant improvement over LoRA without introducing additional parameters relative to LoRA. On the other hand, GMoE's performance does not necessarily improve as $N$ and $k$ increase, and it comes at the cost of significantly increasing the number of parameters. This highlights the efficiency and effectiveness of EMoE compared to GMoE.
>
> We want to take this opportunity to emphasize again the effectiveness of harnessing emergent modularity in pretrained transformers, which entails leveraging the model's inherent structure to yield performance improvements.
>
> > 7. *A final example, which is similarly important for understanding how EMoEs work says "EMoE and dense model only differ in FFNs activations". Surely the EMoE differs by the fact that it is now gated with an enforced sparsity?*
>
> The difference between EMoE and the standard transformer lies in the activation patterns of neurons within the FFNs. In EMoE, the gating mechanism ensures that neurons from experts not selected do not participate in computations, resulting in enforced sparsity, as mentioned by the reviewer.
>
> > 8. *A second note on clarity, I relied on Figure 1 to understand EMoEs as overall Section 3.1 is difficult to parse.*
>
> We guess there might be a misunderstanding on the part of the reviewer. In Section 3.1, our intention was not to introduce the EMoE. The purpose of Section 3.1 was to help readers become familiar with the structure of the FFNs within the transformer and MoE architecture, which would facilitate their understanding of the subsequent operation of decomposing FFNs into MoEs.
>
> > 9. *Having the transpose on the weight matrix $K^T$ just makes it difficult to follow. Additionally, the output of the $x\cdot K^T$ computation is a $d$-dimensional vector ....*
>
> We appreciate the reviewer for pointing out the error in our formula, and we will correct this mistake.
>
> > 10. Also the first usage of $K_i$ does not bold the $i$ but it is bold everywhere else, so that should be corrected (that's not a critique, just wanted to point it out). As a minor extension of this, when would $d$ represent the dimension of the hidden layer and $h$ represent the dimension of the data?
>
> We appreciate the reviewer for pointing out these notation issues, and we will make corrections in the subsequent versions. In our paper, $d$ is the dimension of the hidden layer, and $h$ is the data dimension (also the embedding size).

---

> ### Author Response · Authors · 2023-11-17
> **Response to reviewer qXUa (Part 5)**
>
> > 11. *Finally, while I appreciate that this was aiming to be grounded within the transformer literature , **rephrasing a normal neural network forward pass in terms of keys, queries and values is also just unhelpful.** Especially when the work is inconsistent on what the key is.*
>
> We believe that the comments from the reviewer regarding the inconsistency in defining "key" may arise from a misunderstanding. Throughout our paper, when we refer to "keys," we consistently mean the rows of matrix $K$. We aimed to emphasize the conceptualization of the FFNs' operations as key-value computations, which aids in understanding the decomposition of FFNs into experts. As illustrated in Figure 1(a), if we interpret the first matrix operation in the FFNs as an inner product between the input and keys to obtain activations and the second matrix operation as a weighted sum of values based on these activations, it naturally clusters keys that are jointly activated and combines them with values to form experts, as shown in Figure 1(b). Furthermore, it is worth noting that a body of existing literature [1, 2, 3] views FFNs as key-value memories.
>
> > 12. *Section 3.1 says the keys are the rows of $K$ but then it is more accurate to interpret the centroids of the clusters of $K$ as the keys as this is what is being compared to the query (input) to determine which module to use.*
>
> We believe there might be some misunderstandings here. In Section 3.1, we did not introduce the concept of "module" or delve into the notion of "cluster centroids." The reason for referring to the rows of matrix $K$ as "keys" in Section 3.1 was that we believed that the concept of a "key" could correspond well with the activation of individual neurons. However, it is important to note that the "keys" mentioned here are unrelated to the activation of the "module." The activation related to the "module" is tied to "the average of keys" within the "experts." Furthermore, it is worth mentioning that the "cluster centroids" mentioned by the reviewer could be employed as gate weights in Figure 1(c). However, due to the increased complexity associated with updating them compared to using the average values, we opted for using the average values rather than "centroids.”
>
> > 13. *"Instead, the goal is to demonstrate that dense pre-trained transformers can benefit from emergent modular structures" does not quite summarize accurately what is shown. This would demonstrate that the **emergent modular structure is a phenomenon which lasts in transformers, not some structure which must be algorithmically extracted and enforced.***
>
> Thank you for pointing out the need for clarification. In future versions, we intend to revise the summary: "demonstrate that externalizing the emergent modularity into MoE structures would be beneficial for fine-tuning pre-trained transformers." However, it is necessary to emphasize that EMoE may not represent the sole approach for harnessing emergent modularity. We strongly encourage the research community to explore and propose more advanced methods for externalizing and leveraging emergent modularity.
>
> > 14. *I recommend cutting down on Sections 3.1 and 3.2 as they are not particularly helpful and Figure 1 does a great job expressing the same thing. I also think Section 2 could be shortened for a more in-depth background section focusing on GMoE and LoRA and potentially MoEfication which focuses on how EMoE differs. Finally, a general check of the grammar and vague phrasing is in order.*
>
> We appreciate your suggestions. In our revisions, we will include the high-level ideas we aim to convey in Sections 3.1 and 3.2.  We acknowledge the suggestion to condense Section 2 and instead focus on a more comprehensive background section that delves deeper into GMoE, and MoEfication, emphasizing the distinctions between these approaches and EMoE. We will conduct a thorough grammar and phrasing review to ensure the overall language quality of the manuscript is improved.
>
> [1] Large Memory Layers with Product Keys, https://arxiv.org/abs/1907.05242
>
> [2] Transformer Feed-Forward Layers Build Predictions by Promoting Concepts in the Vocabulary Space, https://arxiv.org/abs/2203.14680
>
> [3] Transformer Feed-Forward Layers Are Key-Value Memories, https://arxiv.org/abs/2012.14913

---

> > ### Comment · Reviewer_qXUa · 2023-11-18
> > **Response to Rebuttal**
> >
> > I thank the authors for their in-depth response.
> >
> > There is a lot to take in here, so I would like to try recenter the discussion on my main concerns. Based on the rebuttal, I see two axis along which EMoE is differentiated from MoEfication and GMoE. The first is at which point fine-tuning occurs relative to the  network split. The second is the manner in which the MoE is formed. For the split point, EMoE and GMoE are split into an MoE before fine-tuning, while MoEfication forms an MoE after fine-tuning. Experts which are split before fine-tuning are call "heterogeneous", while those split after are called "homogeneous" and this is because those split after fine-tuning have all seen the **same** (ie: homo) datasets. For the manner in which the MoE is formed, EMoE and MoEfication split the network itself, while GMoE makes copies of the network to form the experts. Is any of what I have said so far incorrect?
> >
> > Assuming the above is fair (and please do correct me), then I agree with the authors that comparing with GMoE in terms of performance is not fair. But this actually highlights my original main issue, "what is this paper aiming to show"? I see the authors clarified this by saying: "We want to share with the reader that pre-trained transformers exhibit emergent modularities ... and such emergent modularity should be further encouraged at the training stage of the model to achieve the benefit of ID and OOD performance". A comparison with GMoE does not in any way helps address this research question and just serves to confuse as far as I can tell. Even then, the fact that MoEfication can split the hidden layer in a similar manner to EMoE means that indeed at some point in training these modules must have emerged. Thus, this work does not contribute this point. The main point then (again please correct me) that I can see is that there is a benefit to MoEfication prior to fine-tuning (giving the proposed EMoE procedure) rather than using MoEfication after fine-tuning. Is this a fair summary? If this really is the distilled contribution, then it does not come across clearly at all in the current version and I don't think this is a very difficult point to make. That said, the experimental design itself seems to not address this research question, beyond just implying that more specific fine-tuning is helpful. A comparison of fine-tuned EMoE modules with untuned MoEfication modules does not seem fair for determining that the emergent modules before fine-tuning have inherently beneficial properties. A more appropriate experiment to me would a) train both the monolithic networks initially, b) use MoEfication on one network (EMoE model) and not the other (original MoEfication baseline), c) fine-tune both models, d) MoEfication on the baseline (at this point this is all identical to what has been presented), e) instead of evaluation now, both networks are fine-tuned further - controlling for the fact that MoEfication introduces noise which fine-tuning can fix but actually has little to do with the inherent properties of the discover edexpert modules (it is already established that MoEfication introduces noise and so this is not a finding), f) evaluate both networks now.
> >
> > I appreciate that my misunderstanding of the notion of keys may be due to the clarity issues (as I mention in my original review). Thus, I will re-evaluate when the revised version is uploaded. Suffice to say, due to the increased complexity of using average keys for clustering and forming centroids but also using these averages to weight the modules, it is necessary to be precise about notation and the computation involved. Similarly, I will reserve judgement of the new results for the revised paper where a more thorough introduction of the datasets and discussion of the results can be presented. For now, I think the main discussion thread can be around my comments above.

---

> > > ### Author Response · Authors · 2023-11-18
> > > **Response to Response to Rebuttal of reviewer qXUa (Part 1)**
> > >
> > > We appreciate the prompt response from the reviewer, and we are pleased to have addressed the concerns regarding GMoE. Below, we will provide detailed explanations for the other issues raised.
> > >
> > >  > 1. *For the split point, EMoE and GMoE are split into an MoE before fine-tuning, while MoEfication forms an MoE after fine-tuning.*
> > >
> > > We notice that while the reviewer later correctly mentions, "GMoE makes copies of the network to form the experts," we would like to clarify that in the phrase "GMoE are split into an MoE," the term "split" should be replaced with "copy," as we have summarized in the table of Response to reviewer qXUa (part 1).
> > >
> > > > 2. *Experts which are split before fine-tuning are call "heterogeneous", while those split after are called "homogeneous" and this is because those split after fine-tuning have all seen the **same** (ie: homo) datasets.*
> > >
> > > We did not consider "heterogeneous" or "homogeneous" from the perspective of datasets. The use of "heterogeneous" in our context refers to the properties of the weights during the **initialization**. Experts who are split from FFNs are relatively more "heterogeneous" compared to experts copied directly from FFNs, as explained in our response to Question 4 in Response Part 4. "heterogeneous" and "homogeneous" are not directly related to fine-tuning. We emphasize "before fine-tuning" and "after fine-tuning" only to describe the different experimental pipelines for the three methods.
> > >
> > > > 3. *I see the authors clarified this by saying: "We want to share with the reader that pre-trained transformers exhibit emergent modularities ... and such emergent modularity should be further encouraged at the training stage of the model to achieve the benefit of ID and OOD performance". A comparison with GMoE does not in any way helps address this research question and just serves to confuse as far as I can tell ...... The main point then (again please correct me) that I can see is that there is a benefit to MoEfication prior to fine-tuning (giving the proposed EMoE procedure) rather than using MoEfication after fine-tuning. Is this a fair summary?*
> > >
> > > We apologize for the ambiguity in our previous use of "should." We appreciate the reviewer's thorough understanding of our clarified statement and the detailed suggestions based on it. In response to the reviewer's questions, we think the reviewer's focus on our statement, "such emergent modularity should be further encouraged at the training stage," may have centered on the phrase "should ... at training." This interpretation seems to imply that "encourage at the training stage" might be superior to other stages (such as a variant of MoEfication after training). Consequently, the reviewer has summarized our point as favoring "MoEfication before fine-tuning" over "MoEfication after fine-tuning" and has suggested that (1) comparing EMoE only to GMoE is not meaningful because GMoE does not involve emergent modularity, and (2) a comparison should be made between EMoE and an adjusted MoEfication. Given this understanding, we acknowledge that the experimental setup proposed by the reviewer is reasonable **but may not align with our primary research questions.**
> > >
> > > In our previous statement, the focus of "should" was on "encourage the modularity." Alternatively, it could be expressed as "such emergent modularity **can** be further encouraged at the training stage of the model to achieve the benefit of ID and OOD performance." What we intend to convey is that encouraging modularity can bring benefits. Therefore, we primarily compare EMoE with the baseline and the state-of-the-art method GMoE under the same evaluation settings to illustrate the effectiveness of EMoE.
> > >
> > > This conveyed information aligns well with our emphasized research question in the submission:
> > >
> > > *"We explore whether and how standard pre-trained transformers could benefit from emergent modular structures."*
> > >
> > > Furthermore, it is consistent with our response to reviewer xYuL (Part 1):
> > >
> > > *"We want to share the insight and interesting experimental results with the LLM community: Pre-trained Language Models can benefit from their encouraged emergent modularity during the fine-tuning stage."*
> > >
> > > > 4. *That said, the experimental design itself seems to not address this research question, beyond just implying that more specific fine-tuning is helpful.*
> > >
> > > We appreciate the reviewer's feedback. Based on the responses provided in the previous discussion, we believe that comparing EMoE with fine-tuning methods can effectively address our research problem: "Emergent modularity can be further encouraged at the training stage of the model to achieve the benefit of ID and OOD performance." This comparison should prove the effectiveness as long as methods encouraging emergent modularity demonstrate a clear and significant superiority over standard fine-tuning.

---

> > > > ### Author Response · Authors · 2023-11-18
> > > > **Response to Response to Rebuttal of reviewer qXUa (Part 2)**
> > > >
> > > > We hope the previous response contributes to addressing your questions.
> > > >
> > > > Furthermore, we would appreciate your feedback on our responses to your previous concerns:
> > > >
> > > > > 1. *The authors claim that there is a benefit from EMoE and I cannot see how this is consistently the case across Tables 1, 2 and 3. While there is some improvement over the various baselines in each case (ViT, BERT, GPT2 and LoRA) it still appears marginal and fairly inconsistent. This is made worse by the fact that standard deviations are not reported or commented on (while they are referenced in the caption of Table 4.1)*
> > > >
> > > > > 3. …. *Is this addition of complexity worth the benefit over using the vanilla baselines? The answer to that question is likely up to the reader, but failing to mention this or provide the reader with evidence on this appears to be a significant omission.*
> > > >
> > > > > 6. *why are EMoEs given significantly more modules to train (a larger $N$) and use (a larger $K$) than GMoEs? Does this make a comparison unfair?*

---

> > > > ### Comment · Reviewer_qXUa · 2023-11-22
> > > > **Response to Authors**
> > > >
> > > > I would just like to note that, unfortunately, my concerns regarding GMoE have not been addressed - I just agree that using it as a benchmark of performance is not fair.
> > > >
> > > > On the terminology of "heterogeneous" - fair I understand what the authors are saying. I don't believe this was mentioned in such simple terms in the paper and I think it should be.
> > > >
> > > > On the statement: "What we intend to convey is that encouraging modularity can bring benefits" - this is not a sufficient observation in my opinion. But I do believe the authors have unintentionally down-played their contribution with that statement and so I will take it lightly. But my point stands that I don't believe the current experimental design even shows this as there is no control which uses the same exact architecture and training but does not promote modularity. This was my intent when describing an alternative experimental procedure - I believed it made for a more **controlled** experiment. If EMoE beats MoEfication in your experiments it might be due to the final stage of fine-tuning and if EMoE beats GMoE in your experiments it might be due to the "heterogeneity" of the modules and not the modularity itself. Thus, the comparison with these baselines - at least as I see it - do not clearly provide conclusions for the research question.
> > > >
> > > > The statement which the authors make: "Emergent modularity can be further encouraged at the training stage of the model to achieve the benefit of ID and OOD performance." leads me to believe that the authors are saying that the identifiability of modules early in training is the key finding? In the sense that, while MoEfication may find modules at the end of training, it is unclear at what point in training these modules become identifiable. This work shows that in fact these modules are identifiable early on and as a result can be found and then enhanced/promoted more. Is this a fair phrasing then of the authors intended message? But this does not seem to be consistent with the authors other statement in the rebuttal of "Our focus is primarily on investigating the effectiveness of emergent modularity in downstream tasks". Thus I am still unsure.
> > > >
> > > > For the additional feedback questions:
> > > > 1. Have the authors uploaded a new draft of the paper where I can properly consider the experimental results (both new and old)?
> > > > 3. I stand by my original comment that the hyper-parameters need to be spoken about. Even just saying they are not very sensitive - as the authors do in the rebuttal. But a larger discussion would be better obviously and I do think it is prudent.
> > > > 6. Again I think this needs to be spoken about. I now understand the differences between EMoE and GMoE better and see that to begin with comparison between the two is difficult. Not just due to their vastly different hyper-parameters. Again I think a much better background is needed for GMoE.
> > > >
> > > > I am going to still keep my score the same, but will evaluate again if the authors are prepared to continue the discussion.

---

### Author Response · Authors · 2023-11-17
**Genera Response (Part 1 / 3)**

We sincerely appreciate all the reviewers' time and effort in reviewing our paper. We are glad to find that reviewers generally recognize our strengths:
- Our proposed research question: whether the pre-trained Transformer models could benefit from its emergent modularity is **novel [xYuL], insightful [wdQa], significant [qXUa], and potentially insightful [3WYM]**.
- Our empirical studies are **extensive and comprehensive** to support the findings of this paper: emergent modularity could indeed boost the in-distribution and out-of-distribution downstream performance. [**6Xtr, wdQa, 3WYM, qXUa**]

We thank the reviewers for the useful suggestions, which helped improve this paper further. In addition to the point-wise responses to each reviewer, the major additional experimental results and the comparison between other related works that we want to highlight are summarised as follows.

### Comparison with MoE and GMoE

The differences and similarities between our work and the two related works (MoEfication and GMoE) are questioned by reviewers [**xYuL**, **qXUa**]. We highlight their relationship here.
- Similarities: EMoE is derived from MoEfication, as we clarified in our submission. However, a novel method is not the purpose of this work. GMoE is regarded as the main baseline of our paper.  We employ a similar evaluation pipeline. They initially only evaluate the OOD performance on vision tasks. We use its public code to evaluate it on language tasks.
- Differences: **The main difference between the three works lies in motivations and research questions**. **MoEfication** is motivated by the observation that FFNs layers in pre-trained transformers are sparsely activated (many neurons are unused for inputs). It splits FFNs into a sparse MoE, **aiming to reduce the computational cost, further improving inference efficiency**. **GMoE** makes multiple replicates of the original FFN layer and introduces a learned gate to form an MoEs architecture (As their modular structure is introduced by replicated FFNs, modularity within pre-trained FFNs is not explored). They claim that such sparsely-activated architecture could **improve OOD performance** from their theoretical perspective of algorithmic alignment framework [https://openreview.net/forum?id=rJxbJeHFPS]. MoEfication and GMoE do not touch on how emergent modularity influences the training stage.

### Additional Experimental Results

The performance of EMoE is one major concern of the reviewers. Though the performance is not the intended purpose of this work, we supplement extensive experimental results to support our insights (our key contribution): **externalized emergent modularity favors fine-tuning of pre-trained transformers by boosting ID and OOD generalization on various downstream tasks**. Specifically, we supplement two sets of experimental results.

---

> ### Author Response · Authors · 2023-11-17
> **Genera Response (Part 2 / 3)**
>
> 1. A multi-task setting:
>
>     In our analysis (Section 5.1), we have identified that the improvement brought by EMoE is likely associated with mitigating negative transfer. Inspired by this, we choose a multi-task learning setting where negative transfer might be more pronounced. We adopt the codebase of ATTEMPT: Attentional Mixture of Prompt Tuning [https://github.com/AkariAsai/ATTEMPT]. For the in-domain (ID) scenario, we follow the settings outlined in the paper and select six tasks from the Super-GLUE benchmark. For the out-of-domain (OOD) scenario, we take two larger natural language inference (NLI) datasets, MNLI and QNLI, as our ID training data. We subsequently conducted direct testing on four additional NLI datasets from different domains. All hyperparameters unrelated to mixture-of-experts (MoEs) are kept consistent with the baseline, and we have listed the MoEs-related hyperparameters in the following table.
>
>     Our observations are as follows:
>
>     1. EMoE exhibits a substantial improvement compared to the baseline. In the in-domain (ID) setting, the highest improvement reached 7.56, even considering the average performance across the six tasks. In the out-of-domain (OOD) setting, the highest average OOD result across the four datasets improved by 1.58.
>     2. Across various settings of $N$ and $k$, EMoE consistently outperforms the vanilla fone-tuning. Within the hyperparameter search space specified in our paper, EMoE consistently improves at least 2 points over the baseline in the in-domain (ID) setting. This emphasizes the effectiveness of EMoE and EMoE’s robustness to the explored hyperparameter range.
>
>
>
> Table 1: T5-base ID performances.
> |   |   |   |   |   |   |   |   |   |
> |---|---|---|---|---|---|---|---|---|
> |Experts (N)|topk (k)|SuperGLUE-boolq|SuperGLUE-cb|SuperGLUE-wic|SuperGLUE-wsc.fixed|SuperGLUE-rte|SuperGLUE-copa|test avg|
> |Baseline||82.14|85.71|65.83|34.62|74.10|52.00|65.73|
> |8|2|80.67|89.29|65.52|36.54|79.86|56.00|67.98|
> |16|4|81.16|89.29|68.34|51.92|74.10|44.00|68.14|
> |32|8|80.12|78.57|70.85|63.46|82.73|64.00|**73.29**|
> |32|16|81.04|75.00|72.41|57.69|78.42|54.00|69.76|
>
> Table 2: T5-base OOD performances
> |   |   |   |   |   |   |   |   |   |
> |---|---|---|---|---|---|---|---|---|
> |||ID||OOD|||||
> |Experts (N)|topk (k)|mnli|qnli|wnli|rte|superglue-rte|superglue-cb|OOD avg|
> |Baseline||86.2|92.42|50|61.87|62.59|32.14|51.65|
> |8|2|86.27|92.18|50|64.03|62.59|32.14|52.19|
> |16|2|86.44|92.59|52.78|64.03|56.83|39.29|**53.23**|
> |32|8|86.56|92.49|58.33|64.03|61.87|28.57|53.20|

---

> ### Author Response · Authors · 2023-11-17
> **Genera Response (Part 3 / 3)**
>
> 2. An instruction-tuning setting: To further prove that the main conclusion EMoE still holds for larger LLMs, we use the Alpaca dataset to instruction-tune the Llama series models and evaluate it on the MMLU benchmark.
>
>     We have observed the following:
>
>     1. Across model sizes of 7B and 30B, as well as settings such as full-finetuning and Lora tuning, EMoE consistently yields improvements relative to the baseline.
>     2. The choice of $k$ and $N$ proportions remains applicable even in larger-scale models. While variations may be in different settings, they consistently outperform the baseline. This suggests that although additional hyperparameters are introduced, they do not lead to usability challenges.
>
>     There are two observations that we want to share with the reviewers.
>
>     1. We also note that EMoE does not introduce significant computation overhead (refer to wall time and FLOPS)
>     2. Inspired by our analysis results of Section 5.1 (EMoE mainly takes effect during the training stage rather than the inference stage), we recompose EMoE back into the original standard Llama architecture during evaluation, which still achieves substantial gains. We believe that this further enhances the practicality of EMoE because one can only introduce EMoE during fine-tuning and recompose the tuned model into a standard transformer after fine-tuning. We also plan to open-source our codes and implement EMoE into the huggingface for boarder use.
>
> Table 3: Instruction full-tuned Llama2-7B's MMLU scores.
> |   |   |   |   |   |   |
> |---|---|---|---|---|---|
> ||Experts (N)|topk (k)|MMLU score|wall-clock computation times (s)|FLOPS (10^16)|
> |Llama-2-7B|without tuning||46.79|||
> ||full fine-tuning||46.5|4988|8.97|
> ||64|16|**48.08**|5036|9.12|
> ||64|32|47.44|5041|9.24|
>
> Table 4: Instruction LoRA-tuned Llama2-7B's MMLU scores.
> |   |   |   |   |   |   |
> |---|---|---|---|---|---|
> ||Experts (N)|topk (k)|MMLU score|wall-clock computation times (s)|FLOPS (10^17)|
> |Llama-2-7B|without tuning|||||
> ||LoRA-tuning||46.96|1396|6.92|
> ||64|16|**47.58**|1545|7.03|
> ||64|32|47.37|1521|7.13|
>
> Table 5:  Instruction LoRA-tuned Llama-30B's MMLU scores.
> |   |   |   |   |   |   |
> |---|---|---|---|---|---|
> ||Experts (N)|topk (k)|MMLU score|wall-clock computation times (s)|FLOPS (10^18)|
> |Llama-30B|without tuning||51.5|||
> ||LoRA-tuning||56.18|6943|2.25|
> ||256|128|**57.11**|6955|2.25|
> ||256|128|56.64|6974|2.24|
>
>
>
> We will supplement our explanations and additional experimental results in our revision to improve our work (We are still working on it).

---

### Author Response · Authors · 2023-11-21
**update the manuscript**

We have made revisions to the manuscript following the reviewers' requests. The primary refinements were made in sections 1, 2, and 3 to enhance clarity, highlighting the research problem addressed by EMoE and its comparison with related works. Additionally, we have addressed minor points throughout the paper.

Furthermore, we have included results and experimental settings for multi-task learning and instruction tuning settings in the appendix section. We hope these additions address the reviewers' concerns about the significance of EMoE.